# Synthesizing Tasks for Block-based Programming[*]

**Umair Z. Ahmed**[1]    **Maria Christakis**[2]    **Aleksandr Efremov**[2]    **Nigel Fernandez**[2]
**Ahana Ghosh**[2]    **Abhik Roychoudhury**[1]    **Adish Singla**[2]
[1]National University of Singapore, {umair, abhik}@comp.nus.edu.sg,
[2]MPI-SWS, {maria, aefremov, nfernand, gahana, adishs}@mpi-sws.org

## Abstract

Block-based visual programming environments play a critical role in introducing computing concepts to K-12 students. One of the key pedagogical challenges in these environments is in designing new practice tasks for a student that match a desired level of difficulty and exercise specific programming concepts. In this paper, we formalize the problem of synthesizing visual programming tasks. In particular, given a reference visual task $\mathtt{T}^{in}$ and its solution code $\mathtt{C}^{in}$, we propose a novel methodology to automatically generate a set $\{(\mathtt{T}^{out}, \mathtt{C}^{out})\}$ of new tasks along with solution codes such that tasks $\mathtt{T}^{in}$ and $\mathtt{T}^{out}$ are *conceptually similar* but *visually dissimilar*. Our methodology is based on the realization that the mapping from the space of visual tasks to their solution codes is highly discontinuous; hence, directly mutating reference task $\mathtt{T}^{in}$ to generate new tasks is futile. Our task synthesis algorithm operates by first mutating code $\mathtt{C}^{in}$ to obtain a set of codes $\{\mathtt{C}^{out}\}$. Then, the algorithm performs *symbolic execution* over a code $\mathtt{C}^{out}$ to obtain a visual task $\mathtt{T}^{out}$; this step uses the Monte Carlo Tree Search (MCTS) procedure to guide the search in the symbolic tree. We demonstrate the effectiveness of our algorithm through an extensive empirical evaluation and user study on reference tasks taken from the *Hour of Code: Classic Maze* challenge by *Code.org* and the *Intro to Programming with Karel* course by *CodeHS.com*.

## 1   Introduction

Block-based visual programming environments are increasingly used nowadays to introduce computing concepts to novice programmers including children and K-12 students. Led by the success of environments like Scratch [29], initiatives like *Hour of Code* by *Code.org* [24] (HOC) and online platforms like *CodeHS.com* [21], block-based programming has become an integral part of introductory computer science education. Considering HOC alone, over one billion hours of block-based programming activity has been performed so far by over 50 million unique students worldwide [24, 35].

The societal need for enhancing K-12 computing education has led to a surge of interest in developing AI-driven systems for pedagogy of block-based programming [33, 26, 27, 34, 16]. Existing works have studied various aspects of intelligent support, including providing real-time next-step hints when a student is stuck solving a task [20, 36, 18, 17, 9], giving data-driven feedback about a student's misconceptions [31, 19, 28, 30, 35], and demonstrating a worked-out solution for a task when a student lacks the required programming concepts [37]. An underlying assumption when providing such intelligent support is that afterwards the student can practice *new similar tasks* to finally learn the missing concepts. However, this assumption is far from reality in existing systems—the programming tasks are typically hand-curated by experts/tutors, and the available set of tasks is limited. Consider HOC's *Classic Maze* challenge [23], which provides a progression of 20 tasks: Millions of students have attempted these tasks, yet when students fail to solve a task and receive assistance, they cannot practice similar tasks, hindering their ability to master the desired concepts. We seek to tackle this pedagogical challenge by developing techniques for synthesizing new programming tasks.

---

[*]Authors listed alphabetically; Correspondence to: Ahana Ghosh <gahana@mpi-sws.org>.

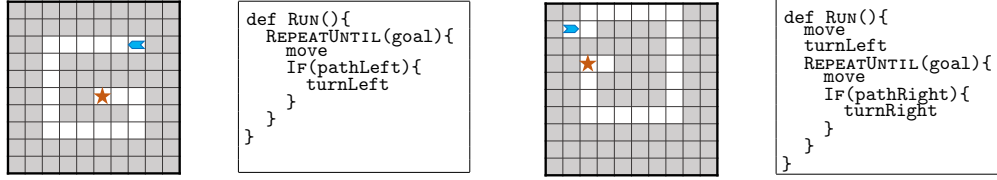

(a) Visual puzzle for $\text{T}^{\text{in}}$  (b) Solution code $\text{C}^{\text{in}}$  (c) Visual puzzle for $\text{T}^{\text{out}}$  (d) Solution code $\text{C}^{\text{out}}$

Figure 1: Illustration of our methodology for task *Maze 16* from the *Hour of Code: Classic Maze* challenge by *Code.org* [23]; the complete list of tasks with their specifications is in Fig. 6.

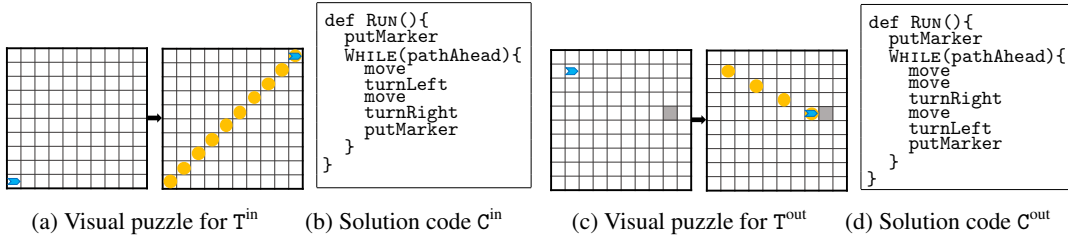

(a) Visual puzzle for $\text{T}^{\text{in}}$  (b) Solution code $\text{C}^{\text{in}}$  (c) Visual puzzle for $\text{T}^{\text{out}}$  (d) Solution code $\text{C}^{\text{out}}$

Figure 2: Illustration of our methodology for task *Diagonal* from the *Intro to Programming with Karel* course by *CodeHS.com* [22]; the complete list of tasks with their specifications is in Fig. 6.

We formalize the problem of synthesizing visual programming tasks of the kind found in popular learning platforms like *Code.org* (see Fig. 1) and *CodeHS.com* (see Fig. 2). As input, we are given a reference task $\text{T}^{\text{in}}$, specified as a visual puzzle, and its solution code $\text{C}^{\text{in}}$. Our goal is to synthesize a set $\{(\text{T}^{\text{out}}, \text{C}^{\text{out}})\}$ of new tasks along with their solution codes that are *conceptually similar* but *visually dissimilar* to the input. This is motivated by the need for practice tasks that on one hand exercise the same concepts, while looking fresh in order to maintain student engagement.

When tackling the problem of synthesizing new tasks with the above desirable properties, three key challenges emerge. First, we are generating problems in a *conceptual domain* with no *well-defined procedure* that students follow to solve a task—consequently, existing work on educational problem generation in procedural domains does not apply in our setting [3, 11]. Second, the mapping from the space of visual tasks to their solution codes is highly discontinuous; hence, template-based problem generation techniques [32, 25] that rely on directly mutating the input to generate new tasks is ineffective (see Section 5 where we use this approach as a baseline). Furthermore, such a direct task-mutation approach would require access to an automated solution synthesizer; however, state-of-the-art program synthesis techniques are not yet on par with experts and their minimal solutions [5, 8, 6]. Third, the space of possible tasks and their solutions is potentially unbounded, and thus, any problem generation technique that relies on exhaustive enumeration is intractable [32, 1, 2].

To overcome these challenges, we propose a novel methodology that operates by first mutating the solution code $\text{C}^{\text{in}}$ to obtain a set of codes $\{\text{C}^{\text{out}}\}$, and then performing symbolic execution over a code $\text{C}^{\text{out}}$ to obtain a visual puzzle $\text{T}^{\text{out}}$. Mutation is efficient by creating an abstract representation of $\text{C}^{\text{in}}$ along with appropriate constraints and querying an SMT solver [4]; any solution to this query is a mutated code $\text{C}^{\text{out}}$. During symbolic execution, we use Monte Carlo Tree Search (MCTS) to guide the search over the (unbounded) symbolic execution tree. We demonstrate the effectiveness of our methodology by performing an extensive empirical evaluation and user study on a set of reference tasks from the *Hour of code* challenge by *Code.org* and the *Intro to Programming with Karel* course by *CodeHS.com*. In summary, our main contributions are:

- We formalize the problem of synthesizing block-based visual programming tasks (Section 2).
- We present a novel approach for generating new visual tasks along with solution codes such that they are conceptually similar but visually dissimilar to a given reference task (Section 3).
- We demonstrate the effectiveness of our approach through an extensive empirical evaluation and user study on reference tasks from real-world programming platforms (Section 4 and Section 5).

## 2  Problem Formulation

**The space of tasks.** We define a task as a tuple $\text{T} := (\text{T}_{\text{vis}}, \text{T}_{\text{store}}, \text{T}_{\text{size}})$, where $\text{T}_{\text{vis}}$ denotes the visual puzzle, $\text{T}_{\text{store}}$ the available block types, and $\text{T}_{\text{size}}$ the maximum number of blocks allowed in the

solution code. For instance, considering the task $T := T^{in}$ in Fig. 1a, $T_{vis}$ is illustrated in Fig. 1a, $T_{store} = \{move, turnL, turnR, \text{REPEATUNTIL}, \text{IF}\}$, and $T_{size} = 4$.

**The space of codes.** The programming environment has a domain-specific language (DSL), which defines the set of valid codes $\mathcal{C}$ and is shown in Fig. 4a. A code $C \in \mathcal{C}$ is characterized by several properties, such as the set $C_{blocks}$ of block types in $C$, the number of blocks $C_{size}$, the depth $C_{depth}$ of the corresponding Abstract Syntax Tree (AST), and the nesting structure $C_{struct}$ representing programming concepts exercised by $C$. For instance, considering the code $C := C^{in}$ in Fig. 1b, $C_{blocks} = \{move, turnL, \text{REPEATUNTIL}, \text{IF}\}$, $C_{size} = 4$, $C_{depth} = 3$, and $C_{struct} = \{\text{RUN}\{\text{REPEATUNTIL}\{\text{IF}\}\}\}$.

Below, we introduce two useful definitions relating the task and code space.

**Definition 1** (Solution code). *$C$ is a solution code for $T$ if the following holds: $C$ successfully solves the visual puzzle $T_{vis}$, $C_{blocks} \subseteq T_{store}$, and $C_{size} \leq T_{size}$. $\mathcal{C}_T$ denotes the set of all solution codes for $T$.*

**Definition 2** (Minimality of a task). *Given a solvable task $T$ with $|\mathcal{C}_T| \geq 1$ and a threshold $\delta \in \mathbb{N}$, the task is minimal if $\nexists C \in \mathcal{C}_T$ such that $C_{size} < T_{size} - \delta$.*

Next, we introduce two definitions formalizing the notion of conceptual similarity. Definition 3 formalizes conceptual similarity of a task $T$ along with one solution code $C$. Since a task can have multiple solution codes, Definition 4 provides a stricter notion of conceptual similarity of a task $T$ for all its solution codes. These definitions are used in our objective of task synthesis in conditions (I) and (V) below.

**Definition 3** (Conceptual similarity of $(T, C)$). *Given a reference $(T^{in}, C^{in})$ and a threshold $\delta \in \mathbb{N}$, a task $T$ along with a solution code $C$ is conceptually similar to $(T^{in}, C^{in})$ if the following holds: $T_{store} = T^{in}_{store}$, $|T_{size} - T^{in}_{size}| \leq \delta$, and $C_{struct} = C^{in}_{struct}$.*

**Definition 4** (Conceptual similarity of $(T, \cdot)$). *Given a reference $(T^{in}, C^{in})$ and a threshold $\delta \in \mathbb{N}$, a task $T$ is conceptually similar to $(T^{in}, C^{in})$ if the following holds: $T_{store} = T^{in}_{store}$, $|T_{size} - T^{in}_{size}| \leq \delta$, and $\forall C \in \mathcal{C}_T$, $C_{struct} = C^{in}_{struct}$.*

**Environment domain knowledge.** We now formalize our domain knowledge about the block-based environment to measure *visual dissimilarity* of two tasks, and capture some notion of *interestingness and quality* of a task. Given tasks $T$ and $T'$, we measure their visual dissimilarity by an environment-specific function $\mathcal{F}_{diss}(T_{vis}, T'_{vis}) \in [0, 1]$. Moreover, we measure generic quality of a task with function $\mathcal{F}_{qual}(T_{vis}, C) \in [0, 1]$. We provide specific instantiations of $\mathcal{F}_{diss}$ and $\mathcal{F}_{qual}$ in our evaluation.

**Objective of task synthesis.** Given a reference task $T^{in}$ and a solution code $C^{in} \in \mathcal{C}_{T^{in}}$ as input, we seek to generate a set $\{(T^{out}, C^{out})\}$ of new tasks along with solution codes that are *conceptually similar* but *visually dissimilar* to the input. Formally, given parameters $(\delta_{size}, \delta_{diss}, \delta_{qual})$, our objective is to synthesize new tasks meeting the following conditions:

(I) $(T^{out}, C^{out})$ is conceptually similar to $(T^{in}, C^{in})$ with threshold $\delta_{size}$ in Definition 3.

(II) $T^{out}$ is visually dissimilar to $T^{in}$ with margin $\delta_{diss}$, i.e., $\mathcal{F}_{diss}(T^{in}_{vis}, T^{out}_{vis}) \geq \delta_{diss}$.

(III) $T^{out}$ has a quality score above threshold $\delta_{qual}$, i.e., $\mathcal{F}_{qual}(T^{out}_{vis}, C^{out}) \geq \delta_{qual}$.

In addition, depending on the use case, it is desirable that the new tasks satisfy the following criteria:

(IV) $C^{out}$ is different from the input solution code, i.e., $C^{out} \neq C^{in}$.

(V) $T^{out}$ is conceptually similar to $(T^{in}, C^{in})$ with threshold $\delta_{size}$ in Definition 4.

(VI) $T^{out}$ is minimal as per Definition 2 for a desired value of $\delta_{mini}$ (e.g., $\delta_{mini} = 0$ or $\delta_{mini} = 1$).

## 3 Our Task Synthesis Algorithm

We now present the pipeline of our algorithm (see Fig. 3), which takes as input a reference task $T^{in}$ and its solution code $C^{in}$, and generates a set $\{(T^{out}, C^{out})\}$ of new tasks with their solution codes. The goal is for this set to be conceptually similar to $(T^{in}, C^{in})$, but for new tasks $\{T^{out}\}$ to

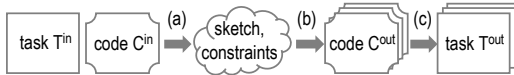

Figure 3: Stages in our task synthesis algorithm.

be visually dissimilar to $T^{in}$. This is achieved by two main stages: (1) mutation of $C^{in}$ to obtain a set $\{C^{out}\}$, and (2) symbolic execution of each $C^{out}$ to create a task $T^{out}$. The first stage, presented in Section 3.1, converts $C^{in}$ into an abstract representation restricted by a set of constraints (Fig. 3(a)), which must be satisfied by any generated $C^{out}$ (Fig. 3(b)). The second stage, described in Section 3.2, applies symbolic execution on each code $C^{out}$ to create a corresponding visual task $T^{out}$ (Fig. 3(c)) while using Monte Carlo Tree Search (MCTS) to guide the search in the symbolic execution tree.

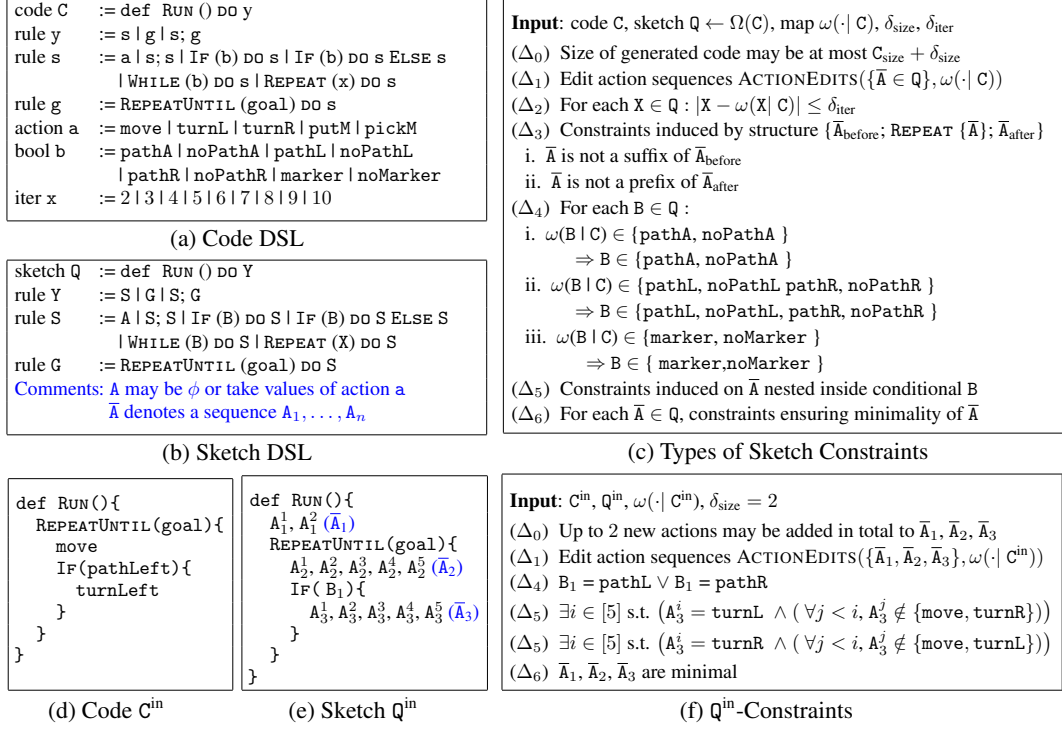

Figure 4: Illustration of key steps in Code Mutation. Fig. 4d shows code $C^{in}$ from Fig. 1b. The code mutation stage, when applied to $C^{in}$, generates many output codes, including $C^{out}$ in Fig. 1d.

## 3.1 Code Mutation

This stage in our pipeline mutates code $C^{in}$ of task $T^{in}$ such that its conceptual elements are preserved. Our mutation procedure consists of three main steps. First, we generate an abstract representation of $C^{in}$, called *sketch*. Second, we restrict the sketch with constraints that describe the space of its concrete instantiations. Although this formulation is inspired from work on generating algebra problems [32], we use it in the entirely different context of generating conceptually similar mutations of $C^{in}$. This is achieved in the last step, where we use the sketch and its constraints to query an SMT solver [4]; the query solutions are mutated codes $\{C^{out}\}$ such that $C^{out}_{struct} = C^{in}_{struct}$ (see Definition 3).

**Step 1: Sketch.** The sketch of code $C$, denoted by $Q$, is an abstraction of $C$ capturing its skeleton and generalizing $C$ to the space of conceptually similar codes. $Q$, expressed in the language of Fig. 4b, is generated from $C$ with mapping $\Omega$. In particular, the map exploits the AST structure of the code: the AST is traversed in a depth-first manner, and all values are replaced with their corresponding sketch variables, i.e., action $a$, bool $b$, and iter $x$ are replaced with $A$, $B$, and $X$, respectively. In the following, we also use mapping $\omega(\cdot \mid C)$, which takes a sketch variable in $Q$ and returns its value in $C$.

In addition to the above, we may extend a variable $A$ to an *action sequence* $\bar{A}$, since any $A$ is allowed to be empty ($\phi$). We may also add an action sequence of length $\delta_{size}$ at the beginning and end of the obtained sketch. As an example, consider the code in Fig. 4d and the resulting sketch in Fig. 4e. Notice that, while we add an action sequence at the beginning of the sketch ($\bar{A}_1$), no action sequence is appended at the end because construct REPEATUNTIL renders any succeeding code unreachable.

**Step 2: Sketch constraints**. Sketch constraints restrict the possible concrete instantiations of a sketch by encoding the required semantics of the mutated codes. All constraint types are in Fig. 4c.

In particular, $\Delta_0$ restricts the size of the mutated code within $\delta_{size}$. $\Delta_1$ specifies the allowed mutations to an action sequence based on its value in the code, given by $\omega(\bar{A} \mid C)$. For instance, this constraint could result in converting all turnLeft actions of a sequence to turnRight. $\Delta_2$ restricts the possible values of the REPEAT counter within threshold $\delta_{iter}$. $\Delta_3$ ensures that the REPEAT counter is optimal, i.e., action subsequences before and after this construct are not nested in it. $\Delta_4$ specifies the possible values of the IF condition based on its value in the code, given by $\omega(B \mid C)$. $\Delta_5$ refers to constraints imposed on action sequences nested within conditionals. As an example, consider

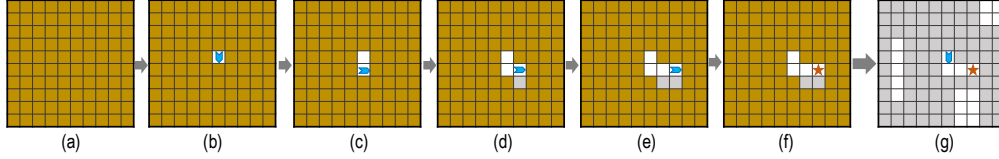

Figure 5: Illustration of symbolic execution on $C^{out}$ from Fig. 1d. (b) shows the initial configuration of the agent's location and orientation as well as the status of the grid cells (*unknown, free, blocked, goal*). (c)–(e) show the symbolic execution steps where conditions `goal` and `pathRight` are False. (f) shows the step where `goal` is True. (g) shows the post-processing step where a puzzle $T^{out}_{vis}$ is obtained.

$\Delta_5$ in Fig. 4f, which states that if $B_1$ = `pathLeft`, then the nested action sequence must have at least one `turnLeft` action, and the first occurrence of this action must not be preceded by a `move` or `turnRight`, thus preventing invalid actions within the conditional. $\Delta_6$ ensures minimality of an action sequence, i.e., optimality of the constituent actions to obtain the desired output. This constraint would, for instance, eliminate redundant sequences such as `turnLeft`, `turnRight`, which does not affect the output, or `turnLeft`, `turnLeft`, `turnLeft`, whose output could be achieved by a single `turnRight`. All employed elimination sequences can be found in the supplementary material. The entire list of constraints applied on the solution code in Fig. 4d is shown in Fig. 4f.

**Step 3: SMT query.** For a sketch Q generated from code C and its constraints, we pose the following query to an SMT solver: (sketch Q, Q-constraints). As a result, the solver generates a set of instantiations, which are conceptually similar to C. In our implementation, we used the Z3 solver [7]. For the code in Fig. 4d, Z3 generated 66 mutated codes in 0.8s from an exhaustive space of 2, 997 possible codes with $\delta_{size} = 2$. One such mutation is shown in Fig. 1d.

While this approach generates codes that are devoid of most semantic irregularities, it has its limitations. Certain irregularities continue to exist in some generated codes: An example of such a code included the action sequence `move`, `turnLeft`, `move`, `turnLeft`, `move`, `turnLeft`, `move`, `turnLeft`, which results in the agent circling back to its initial location in the task space. This kind of undesirable behaviour is eliminated in the symbolic execution stage of our pipeline.

### 3.2 Symbolic Execution

Symbolic execution [13] is an automated test-generation technique that symbolically explores execution paths in a program. During exploration of a path, it gathers symbolic constraints over program inputs from statements along the path. These constraints are then mutated (according to a search strategy), and an SMT solver is queried to generate new inputs that explore another path.

**Obtaining visual tasks with symbolic execution.** This stage in our pipeline applies symbolic execution on each generated code $C^{out}$ to obtain a suitable visual task $T^{out}$. The program inputs of $C^{out}$ are the agent's initial location/orientation and the status of the grid cells (*unknown, free, blocked, marker, goal*), which is initially *unknown*. Symbolic execution collects constraints over these from code statements. As in Fig. 5 for one path, symbolic execution generates a visual task for each path in $C^{out}$.

However, not all of these tasks are suitable. For instance, if the goal is reached after the first `move` in Fig. 1d, all other statements in $C^{out}$ are not covered, rendering the task less suitable for this code. Naïvely, symbolic execution could first enumerate all paths in $C^{out}$ and their corresponding tasks, and then rank them in terms of suitability. However, solution codes may have an unbounded number of paths, which leads to *path explosion*, that is, the inability to cover all paths with tractable resources.

**Guiding symbolic execution using Monte Carlo Tree Search (MCTS).** To address this issue, we use MCTS [14] as a search strategy in symbolic execution with the goal of generating more suitable tasks with fewer resources—we define task suitability next. Symbolic execution has been previously combined with MCTS in order to direct the exploration towards costly paths [15]. In the supplementary material, we provide an example demonstrating how MCTS could guide the symbolic execution in generating more suitable tasks.

As previously observed [12], a critical component of effectively applying MCTS is to define an evaluation function that describes the desired properties of the output, i.e., the visual tasks. Tailoring the evaluation function to our unique setting is exactly what differentiates our approach from existing work. In particular, our evaluation function, $\mathcal{F}_{score}$, distinguishes suitable tasks by assigning a score ($\in [0, 1]$) to them, which guides the MCTS search. A higher $\mathcal{F}_{score}$ indicates a more suitable task.

| Task T | $T_{store}$ | $T_{size}$ (= $C_{size}$) | $C_{depth}$ | Type: Source |
|---|---|---|---|---|
| H1 | move, turnL, turnR | 5 | 1 | HOC: *Maze 4* [23] |
| H2 | move, turnL, turnR, REPEAT | 3 | 2 | HOC: *Maze 7* [23] |
| H3 | move, turnL, turnR, REPEAT | 5 | 2 | HOC: *Maze 8* [23] |
| H4 | move, turnL, turnR, REPEATUNTIL | 5 | 2 | HOC: *Maze 12* [23] |
| H5 | move, turnL, turnR, REPEATUNTIL, IF | 4 | 3 | HOC: *Maze 16* [23] |
| H6 | move, turnL, turnR, REPEATUNTIL, IFELSE | 4 | 3 | HOC: *Maze 18* [23] |
| K7 | move, turnL, turnR, pickM, putM | 5 | 1 | Karel: *Our first* [22] |
| K8 | move, turnL, turnR, pickM, putM, REPEAT | 4 | 2 | Karel: *Square* [22] |
| K9 | move, turnL, turnR, pickM, putM, REPEAT, IFELSE | 5 | 3 | Karel: *One ball in each spot* [22] |
| K10 | move, turnL, turnR, pickM, putM, WHILE | 7 | 2 | Karel: *Diagonal* [22] |

Figure 6: Datasets for HOC and Karel tasks.

Its constituent components are: (i) $\mathcal{F}_{cov}(T_{vis}^{out}, C^{out}) \in \{0, 1\}$, which evaluates to 1 in the event of complete coverage of code $C^{out}$ by task $T_{vis}^{out}$ and 0 otherwise; (ii) $\mathcal{F}_{diss}(T_{vis}^{out}, T_{vis}^{in}) \in [0, 1]$, which evaluates the dissimilarity of $T^{out}$ to $T^{in}$ (see Section 2); (iii) $\mathcal{F}_{qual}(T_{vis}^{out}, C^{out}) \in [0, 1]$, which evaluates the quality and validity of $T^{out}$; (iv) $\mathcal{F}_{nocrash}(T_{vis}^{out}, C^{out}) \in \{0, 1\}$, which evaluates to 0 in case the agent crashes into a wall and 1 otherwise; and (v) $\mathcal{F}_{nocut}(T_{vis}^{out}, C^{out}) \in \{0, 1\}$, which evaluates to 0 if there is a *shortcut sequence* of actions (a in Fig. 4a) smaller than $C_{size}^{out}$ that solves $T^{out}$ and 1 otherwise. $\mathcal{F}_{qual}$ and $\mathcal{F}_{nocut}$ also resolve the limitations of our mutation stage by eliminating codes and tasks that lead to undesirable agent behavior. We instantiate $\mathcal{F}_{score}$ in the next section.

## 4 Experimental Evaluation

In this section, we evaluate our task synthesis algorithm on HOC and Karel tasks. Our implementation is publicly available.[2] While we give an overview of key results here, a detailed description of our setup and additional experiments can be found in the supplementary material.

### 4.1 Reference Tasks and Specifications

**Reference tasks.** We use a set of ten reference tasks from HOC and Karel, shown in Fig. 6. The HOC tasks were selected from the *Hour of Code: Classic Maze* challenge by *Code.org* [23] and the Karel tasks from the *Intro to Programming with Karel* course by *CodeHS.com* [22]. The DSL of Fig. 4a is generic in that it includes both HOC and Karel codes, with the following differences: (i) construct WHILE, marker-related actions putM, pickM, and conditions noPathA, noPathL, noPathR, marker, noMarker are specific to Karel only; (ii) construct REPEATUNTIL and goal are specific to HOC only. Furthermore, the puzzles for HOC and Karel are of different styles (see Fig. 1 and Fig. 2). For all tasks, the grid size of the puzzles is fixed to $10 \times 10$ cells (grid-size parameter $n = 10$).

**Specification of scoring functions.** $\mathcal{F}_{qual}(T_{vis}^{out}, C^{out}) \in [0, 1]$ was approximated as the sum of the normalized counts of 'moves', 'turns', 'segments', and 'long-segments' in the grid; segments and long-segments are sequences of $\geq 3$ and $\geq 5$ move actions respectively. More precisely, for HOC tasks, we used the following function where features are computed by executing $C^{out}$ on $T_{vis}^{out}$:

$$\mathcal{F}_{qual}^{HOC}(T_{vis}^{out}, C^{out}) = \frac{1}{4}\Big(\frac{\#\text{moves}}{2n} + \frac{\#\text{turns}}{n} + \frac{\#\text{segments}}{n/2} + \frac{\#\text{long-segments}}{n/3}\Big).$$

Furthermore, in our implementation, $\mathcal{F}_{qual}(\cdot)$ value was set to 0 when $\mathcal{F}_{nocrash}(\cdot) = 0$. For Karel tasks, $\mathcal{F}_{qual}$ additionally included the normalized counts of putM and pickM, and is provided in the supplementary material. $\mathcal{F}_{diss}(T_{vis}^{out}, T_{vis}^{in}) \in [0, 1]$ was computed based on the dissimilarity of the agent's initial location/orientation w.r.t. $T_{vis}^{in}$, and the grid-cell level dissimilarity based on the Hamming distance between $T_{vis}^{out}$ and $T_{vis}^{in}$. More precisely, we used the following function:

$$\mathcal{F}_{diss}(T_{vis}^{out}, T_{vis}^{in}) = \frac{1}{3}\Big(\text{diss}(\text{loc} \mid T_{vis}^{out}, T_{vis}^{in}) + \text{diss}(\text{dir} \mid T_{vis}^{out}, T_{vis}^{in}) + \text{diss}(\text{grid-cells} \mid T_{vis}^{out}, T_{vis}^{in})\Big)$$

where $\text{diss}(\text{loc} \mid T_{vis}^{out}, T_{vis}^{in}) \in \{0, 1\}$, $\text{diss}(\text{dir} \mid T_{vis}^{out}, T_{vis}^{in}) \in \{0, 1\}$, and $\text{diss}(\text{grid-cells} \mid T_{vis}^{out}, T_{vis}^{in}) \in [0, 1]$ (after the Hamming distance is normalized with a factor of $\frac{2}{n^2}$).

| Task $\mathtt{T}^{\text{in}}$ | Code Mutation | | | | Symbolic Execution | | | Fraction of $\mathtt{T}^{\text{out}}$ with criteria | | |
|---|---|---|---|---|---|---|---|---|---|---|
| | $2{:}\#\mathtt{C}^{\text{out}}_{\Delta=0}$ | $3{:}\#\mathtt{C}^{\text{out}}_{\Delta=0,1}$ | $4{:}\#\mathtt{C}^{\text{out}}_{\Delta=\text{all}}$ | $5{:}\text{Time}$ | $6{:}\#\mathtt{C}^{\text{out}}$ | $7{:}\#\mathtt{T}^{\text{out}}$ | $8{:}\text{Time}$ | $9{:}\text{(V)}$ | $10{:}\text{(VI)}_{\delta_{\text{mini}}=1}$ | $11{:}\text{(VI)}_{\delta_{\text{mini}}=0}$ |
| H1 | $3,159$ | 112 | 64 | 0.6s | 28 | 272 | 68s | 1.00 | 1.00 | 1.00 |
| H2 | $8,991$ | 594 | 138 | 1.7s | 48 | 428 | 61s | 1.00 | 1.00 | 1.00 |
| H3 | $798,255$ | $13,122$ | 720 | 13.3s | 196 | $1,126$ | 60s | 0.90 | 0.98 | 0.90 |
| H4 | $5,913$ | 152 | 108 | 1.0s | 44 | 404 | 167s | 1.00 | 1.00 | 0.50 |
| H5 | $2,997$ | 294 | 66 | 0.8s | 46 | 444 | 348s | 0.98 | 0.59 | 0.27 |
| H6 | $1,728$ | 294 | 54 | 0.6s | 48 | 480 | 347s | 0.80 | 0.45 | 0.07 |
| K7 | $96,875$ | 150 | 122 | 1.3s | 122 | $1,196$ | 61s | 1.00 | 1.00 | 1.00 |
| K8 | $484,875$ | $4,506$ | 990 | 11.6s | 469 | $4,506$ | 63s | 1.00 | 1.00 | 1.00 |
| K9 | $8.595 \times 10^6$ | $60,768$ | 888 | 11.3s | 432 | $4,258$ | 185s | 0.92 | 0.92 | 0.88 |
| K10 | $132.625 \times 10^6$ | $19,328$ | $1,404$ | 17.1s | 532 | $5,032$ | 158s | 1.00 | 1.00 | 1.00 |

Figure 7: Results on HOC and Karel tasks; details are provided in Section 4.

Next, we define the evaluation function $\mathcal{F}_{\text{score}}(\mathtt{T}^{\text{out}}, \mathtt{C}^{\text{out}}, \mathtt{T}^{\text{in}}, \mathtt{C}^{\text{in}}) \in [0, 1]$ used by MCTS:

$$\mathcal{F}_{\text{score}}(\mathtt{T}^{\text{out}}, \mathtt{C}^{\text{out}}, \mathtt{T}^{\text{in}}, \mathtt{C}^{\text{in}}) = \underbrace{\mathbb{1}\Big(\mathcal{F}_{\text{qual}}(\mathtt{T}^{\text{out}}_{\text{vis}}, \mathtt{C}^{\text{out}}) \geq \delta_{\text{qual}}, \mathcal{F}_{\text{nocrash}}(\mathtt{T}^{\text{out}}_{\text{vis}}, \mathtt{C}^{\text{out}}) = 1, \mathcal{F}_{\text{nocut}}(\mathtt{T}^{\text{out}}_{\text{vis}}, \mathtt{C}^{\text{out}}) = 1\Big)}_{(i)} \cdot$$

$$\underbrace{\Big[\alpha_1 \mathcal{F}_{\text{cov}}(\mathtt{T}^{\text{out}}_{\text{vis}}, \mathtt{C}^{\text{out}}) + \alpha_2 \mathcal{F}_{\text{qual}}(\mathtt{T}^{\text{out}}_{\text{vis}}, \mathtt{C}^{\text{out}}) + \alpha_3 \mathcal{F}_{\text{diss}}(\mathtt{T}^{\text{out}}_{\text{vis}}, \mathtt{T}^{\text{in}}_{\text{vis}})\Big]}_{(ii)}$$

where $\mathbb{1}$ is an indicator function and each constant $\alpha = 1/3$. Component (ii) in the above function supplies the gradients for guiding the search in MCTS; Component (i) is applied at the end of the MCTS run to pick the output. More precisely, the *best* task (i.e, the one with the highest $\mathcal{F}_{\text{score}}$ value) is picked only from the pool of generated tasks which have $\mathcal{F}_{\text{score}}(\cdot) > 0$ and satisfy $\mathcal{F}_{\text{cov}}(\cdot) = 1$.

**Specification of task synthesis and MCTS.** As per Section 2, we set the following thresholds for our algorithm: (i) $\delta_{\text{size}} = 2$, (ii) $\delta_{\text{diss}} = 0.33$, and (iii) $\delta_{\text{qual}} = 0.2$ for codes with WHILE or REPEATUNTIL, and $0.05$ otherwise. We run MCTS 10 times per code, with each run generating one task. We set the maximum iterations of a run to 2 million (M) and the exploration constant to 2 [14]. Even when considering a tree depth of $2n\ (= 20)$, there are millions of leaves for difficult tasks H5 and H6, reflecting the complexity of task generation. For each code $\mathtt{C}^{\text{out}}$, we generated 10 different visual tasks. To ensure sufficient diversity among the tasks generated for the same code, we introduced a measure $\mathcal{F}_{\text{diversity}}$. This measure, not only ensures visual task dissimilarity, but also ensures sufficient diversity in entire symbolic paths during generation (for details, see supplementary material).

## 4.2 Results

**Performance of task synthesis algorithm.** Fig. 7 shows the results of our algorithm. The second column illustrates the enormity of the unconstrained space of mutated codes; we only impose size constraint $\Delta_0$ from Fig. 4c. We then additionally impose constraint $\Delta_1$ resulting in a partially constrained space of mutated codes (column 3), and finally apply all constraints from Fig. 4c to obtain the final set of generated codes (column 4). This reflects the systematic reduction in the space of mutated codes by our constraints. Column 5 shows the total running time for generating the final codes, which denotes the time taken by Z3 to compute solutions to our mutation query. As discussed in Section 3.1, few codes with semantic irregularities still remain after the mutation stage. The symbolic execution stage eliminates these to obtain the reduced set of valid codes (column 6). Column 7 shows the final number of generated tasks and column 8 is the average time per output task (i.e., one MCTS run).

**Analyzing output tasks.** We further analyze the generated tasks based on the objectives of Section 2. All tasks satisfy properties (I)–(III) by design. Objective (IV) is easily achieved by excluding generated tasks for which $\mathtt{C}^{\text{out}} = \mathtt{C}^{\text{in}}$. For a random sample of $100$ of the generated tasks per reference task, we performed manual validation to determine whether objectives (V) and (VI) are met. The fraction of tasks that satisfy these objectives is listed in the last three columns of Fig. 7. We observe that the vast majority of tasks meet the objectives, even if not by design. For H6, the fraction of tasks satisfying (VI) is low because the corresponding codes are generic enough to solve several puzzles.

**Deep dive into an MCTS run.** To offer more insight into the task generation process, we take a closer look at an MCTS run for task H5, shown in Fig. 8. Fig. 8a illustrates the improvement in various components of $\mathcal{F}_{\text{score}}$ as the number of MCTS iterations increases. Best tasks at different iterations are shown in Fig. 8b, 8c, 8d. As expected, the more the iterations, the better the tasks are.

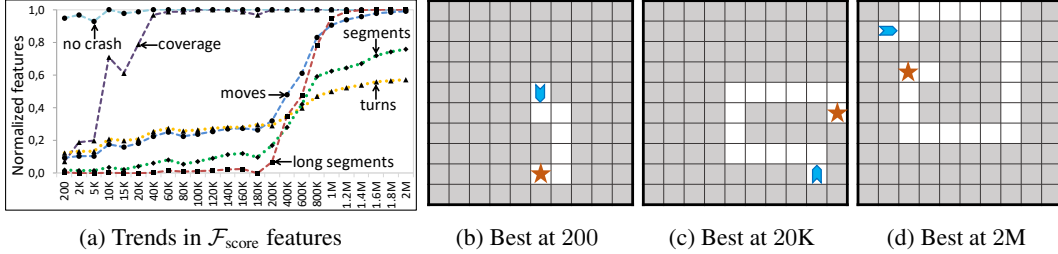

| (a) Trends in $\mathcal{F}_{\text{score}}$ features | (b) Best at 200 | (c) Best at 20K | (d) Best at 2M |

Figure 8: Illustration of a single MCTS run on $\mathtt{C}^{\text{out}}$ from Fig. 1d obtained from solution code of task H5 by mutation. (a) shows the temporal trends of different feature values in $\mathcal{F}_{\text{score}}$ averaged over a time window of 100 steps. (b)–(d) show the best, i.e., highest scoring, tasks generated up to times $2 \times 10^2$, $2 \times 10^4$, and $2 \times 10^6$ respectively. $\mathtt{T}^{\text{out}}_{\text{vis}}$ shown in Fig. 1c is the puzzle produced in (d).

**Remarks.** We also ran the mutation stage by enumerating the programs within size constraints and then post-checking other constraints without Z3. This implementation leads to a run-time increase by a factor of 10 to 100 for different tasks. So, Z3 seems to be very effective by jointly considering all the constraints. As a search method, although MCTS seems computationally expensive, the actual run-time and memory footprint of an MCTS run depend on the unique traces explored (i.e., unique symbolic executions done)—this number is typically much lower than the number of iterations, also see discussion in the supplementary material. Considering the MCTS output in Figs. 8c, 8d, to obtain a comparable evaluation score through a random search, the corresponding number of unique symbolic executions required is at least 10 times more than executed by MCTS. We note that while we considered one I/O pair for Karel tasks, our methodology can be easily extended to multiple I/O pairs by adapting techniques designed for generating diverse tasks.

## 5 User Study and Comparison with Alternate Methods

In this section, we evaluate our task synthesis algorithm with a user study focusing on tasks H2, H4, H5, and H6. We developed an online app[3], which uses the publicly available toolkit of Blockly Games [10] and provides an interface for a participant to practice block-based programming tasks for HOC. Each "practice session" of the study involves three steps: (i) a reference task $\mathtt{T}^{\text{in}} \in \{\mathtt{H2}, \mathtt{H4}, \mathtt{H5}, \mathtt{H6}\}$ is shown to the participant along with its solution code $\mathtt{C}^{\text{in}}$, (ii) a new task $\mathtt{T}^{\text{out}}$ is generated for which the participant has to provide a solution code, and (iii) a post-survey asks the participant to assess the visual dissimilarity of the two tasks on a 4-point Likert scale as used in [25]. Details on the app interface and questionnaire are provided in the supplementary material. Participants for the study were recruited through Amazon Mechanical Turk. We only selected four tasks due to the high cost involved in conducting the study (about 1.8 USD per participant). The number of participants and their performance are documented in Fig. 9.

**Baselines and methods evaluated.** We evaluated four different methods, including three baselines (SAME, TUTOR, MUTTASK) and our algorithm (SYNTASK). SAME generates tasks such that $\mathtt{T}^{\text{in}} = \mathtt{T}^{\text{out}}$. TUTOR produces tasks that are similar to $\mathtt{T}^{\text{in}}$ and designed by an expert. We picked similar problems from the set of 20 *Classic Maze* challenge [23] tasks exercising the same programming concepts: *Maze 6, 9* for H2, *Maze 11, 13* for H4, *Maze 15, 17* for H5, and *Maze 19* for H6.

MUTTASK generated tasks by directly mutating the grid-world of the original task, i.e., by moving the agent or goal by up to two cells and potentially changing the agent's orientation. A total of 18, 20, 15, and 17 tasks were generated for H2, H4, H5, and H6, respectively. Fig. 10 shows two output tasks for H4 and illustrates the challenge in directly mutating the input task, given the high discontinuity in mapping from the space of tasks to their codes. For H4, a total of 14 out of 20 new tasks were structurally very different from the input.

SYNTASK uses our algorithm to generate tasks. We picked the generated tasks from three groups based on the size of the code mutations from which they were produced, differing from the reference solution code by $+\delta_{\text{size}}$ for $\delta_{\text{size}} \in \{0, 1, 2\}$. For H2 and H4, we randomly selected 5 tasks from each group, for a total of 15 new tasks per reference task. For H5 and H6, we selected 10 tasks from the first group ($\delta_{\text{size}} = 0$) only, due to their complexity stemming from nested constructs in their codes. We observed that TUTOR tasks for H5, H6 were also of $\delta_{\text{size}} = 0$, i.e., $\mathtt{C}^{\text{out}}_{\text{size}} = \mathtt{C}^{\text{in}}_{\text{size}}$. All the generated tasks picked for SYNTASK adhere to properties (I)–(VI) in Section 2.

| Method | Total participants | | | | | Fraction of tasks solved | | | | | Time spent in secs | | | | | Visual dissimilarity | | | | |
|---|---|---|---|---|---|---|---|---|---|---|---|---|---|---|---|---|---|---|---|---|
| | **H-** | H2 | H4 | H5 | H6 | **H-** | H2 | H4 | H5 | H6 | **H-** | H2 | H4 | H5 | H6 | **H-** | H2 | H4 | H5 | H6 |
| SAME | **96** | 24 | 24 | 24 | 24 | **.94** | .92 | 1.00 | .96 | .88 | **89** | 60 | 59 | 93 | 145 | **1.07** | 1.12 | 1.04 | 1.00 | 1.12 |
| TUTOR | **170** | 48 | 48 | 49 | 25 | **.90** | .90 | .92 | .88 | .92 | **121** | 107 | 113 | 118 | 169 | **2.90** | 2.81 | 2.79 | 2.96 | 3.16 |
| MUTTASK | **278** | 72 | 79 | 60 | 67 | **.68** | .76 | .71 | .65 | .60 | **219** | 135 | 299 | 219 | 215 | **2.17** | 2.36 | 2.33 | 1.95 | 1.99 |
| SYNTASK | **197** | 59 | 57 | 40 | 41 | **.89** | .92 | .89 | .92 | .83 | **144** | 85 | 183 | 130 | 189 | **2.63** | 2.41 | 2.42 | 2.68 | 3.20 |

Figure 9: User study results for HOC tasks (**H-**represents all tasks in the study); see Section 5.

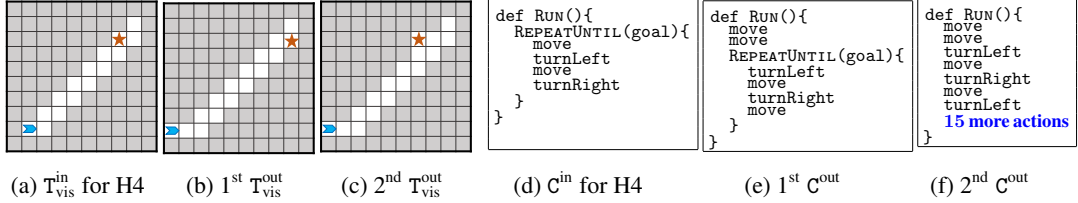

(a) $T_{vis}^{in}$ for H4    (b) $1^{st}$ $T_{vis}^{out}$    (c) $2^{nd}$ $T_{vis}^{out}$    (d) $C^{in}$ for H4    (e) $1^{st}$ $C^{out}$    (f) $2^{nd}$ $C^{out}$

Figure 10: MUTTASK applied to H4. $T_{vis}^{in}$ and $C^{in}$ are shown in (a) and (d). (b)–(c) illustrate two tasks $T_{vis}^{out}$ obtained via small mutations of $T_{vis}^{in}$. (e) is the smallest solution code for (b) and is structurally similar to $C^{in}$. (f) is the smallest solution code for (c) and is drastically different from $C^{in}$.

**Results on task solving.** In terms of successfully solving the generated tasks, SAME performed best (mean success = 0.94) in comparison to TUTOR (mean = 0.90), SYNTASK (mean = 0.89), and MUTTASK (mean = 0.68)—this is expected given the tasks generated by SAME. In comparison to TUTOR, the performance of SYNTASK was not significantly different ($\chi^2 = 0.04, p = 0.83$); in comparison to MUTTASK, SYNTASK performed significantly better ($\chi^2 = 28.74, p < e^{-8}$). The complexity of the generated tasks is also reflected in the average time that participants spent on solving them. As shown in Fig. 9, they spent more time solving the tasks generated by MUTTASK.

**Results on visual task dissimilarity.** Visual dissimilarity was measured on a Likert scale ranging from 1–4, 1 being highly similar and 4 highly dissimilar. Comparing the dissimilarity of the generated tasks w.r.t. the reference task, we found that the performance of SAME was worst (mean dissimilarity = 1.07), while that of TUTOR was best (mean = 2.90). SYNTASK (mean = 2.63) performed significantly better than MUTTASK (mean = 2.17), yet slightly worse than TUTOR. This is because TUTOR generates tasks with additional distracting paths and noise, which can also be done by our algorithm (although not done for this study). Moreover, for H2, which had no conditionals, the resulting codes were somewhat similar, and so were the generated puzzles. When excluding H2 from the analysis, the difference between SYNTASK (mean = 2.72) and TUTOR (mean =2.93) was not statistically significant. A detailed distribution of the responses can be found in the supplementary material.

**Remarks.** SAME's performance in terms of tasks solved is below 1.00, possibly because participants overlooked the solution of Step 1, unaware they will be receiving the same task in Step 2, and the app did not allow them to go back to Step 1. This user study provides a proof-of-concept; more elaborate studies are needed to fully reach the motivational goal of teaching K-12 students, and evaluate the long term impact on students' concept learning. As additional studies, it would be important to understand the sensitivity of user study results w.r.t. the Likert scale definition; another possibility is to use pairwise comparisons in eliciting user evaluations.

## 6 Conclusions and Outlook

We developed techniques for a critical aspect of pedagogy in block-based programming: Automatically generating new tasks that exercise specific programming concepts, while looking visually dissimilar to input. We demonstrated the effectiveness of our methodology through an extensive empirical evaluation and user study on reference tasks from popular programming platforms. We believe our techniques have the potential to drastically improve the success of pedagogy in block-based visual programming environments by providing tutors and students with a substantial pool of new tasks. Beyond the application domain of programming education, our methodology can be used for generating large-scale datasets consisting of tasks and solution codes with desirable characteristics—this can be potentially useful for training neural program synthesis methods.

There are several promising directions for future work, including but not limited to: Learning a policy to guide the MCTS procedure (instead of running vanilla MCTS); automatically learning the constraints and cost function from a human-generated pool of problems; and applying our methodology to other programming environments (e.g., Python problems).

## Broader Impact

This paper develops new techniques for improving pedagogy in block-based visual programming environments. Such programming environments are increasingly used nowadays to introduce computing concepts to novice programmers, and our work is motivated by the clear societal need of enhancing K-12 computing education. In existing systems, the programming tasks are hand-curated by tutors, and the available set of tasks is typically very limited. This severely limits the utility of existing systems for long-term learning as students do not have access to practice tasks for mastering the programming concepts.

We take a step towards tackling this challenge by developing a methodology to generate new practice tasks for a student that match a desired level of difficulty and exercise specific programming concepts. Our task synthesis algorithm is able to generate 1000's of new similar tasks for reference tasks taken from the *Hour of Code: Classic Maze* challenge by *Code.org* and the *Intro to Programming with Karel* course by *CodeHS.com*. Our extensive experiments and user study further validate the quality of the generated tasks. Our task synthesis algorithm could be useful in many different ways in practical systems. For instance, tutors can assign new practice tasks as homework or quizzes to students to check their knowledge, students can automatically obtain new similar tasks after they failed to solve a given task and received assistance, and intelligent tutoring systems could automatically generate a personalized curriculum of problems for a student for long-term learning.

## Acknowledgments and Disclosure of Funding

We would like to thank the anonymous reviewers for their helpful comments. Ahana Ghosh was supported by Microsoft Research through its PhD Scholarship Programme. Umair Z. Ahmed and Abhik Roychoudhury were supported by the National Research Foundation, Singapore and National University of Singapore through its National Satellite of Excellence in Trustworthy Software Systems (NSOE-TSS) project under the National Cybersecurity R&D (NCR) Grant award no. NRF2018NCR-NSOE003-0001.

## Footnotes

[2]https://github.com/adishs/neurips2020_synthesizing-tasks_code

[3]https://www.teaching-blocks.cc/

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
