[Supplementary Material]

# Synthesizing Tasks for Block-based Programming[*]

**Umair Z. Ahmed**[1]    **Maria Christakis**[2]    **Aleksandr Efremov**[2]    **Nigel Fernandez**[2]
**Ahana Ghosh**[2]    **Abhik Roychoudhury**[1]    **Adish Singla**[2]
[1]National University of Singapore, {umair, abhik}@comp.nus.edu.sg,
[2]MPI-SWS, {maria, aefremov, nfernand, gahana, adishs}@mpi-sws.org

## Abstract

Block-based visual programming environments play a critical role in introducing computing concepts to K-12 students. One of the key pedagogical challenges in these environments is in designing new practice tasks for a student that match a desired level of difficulty and exercise specific programming concepts. In this paper, we formalize the problem of synthesizing visual programming tasks. In particular, given a reference visual task $T^{in}$ and its solution code $C^{in}$, we propose a novel methodology to automatically generate a set $\{(T^{out}, C^{out})\}$ of new tasks along with solution codes such that tasks $T^{in}$ and $T^{out}$ are *conceptually similar* but *visually dissimilar*. Our methodology is based on the realization that the mapping from the space of visual tasks to their solution codes is highly discontinuous; hence, directly mutating reference task $T^{in}$ to generate new tasks is futile. Our task synthesis algorithm operates by first mutating code $C^{in}$ to obtain a set of codes $\{C^{out}\}$. Then, the algorithm performs *symbolic execution* over a code $C^{out}$ to obtain a visual task $T^{out}$; this step uses the Monte Carlo Tree Search (MCTS) procedure to guide the search in the symbolic tree. We demonstrate the effectiveness of our algorithm through an extensive empirical evaluation and user study on reference tasks taken from the *Hour of Code: Classic Maze* challenge by *Code.org* and the *Intro to Programming with Karel* course by *CodeHS.com*.

## 1   Introduction

Block-based visual programming environments are increasingly used nowadays to introduce computing concepts to novice programmers including children and K-12 students. Led by the success of environments like Scratch [29], initiatives like *Hour of Code* by *Code.org* [24] (HOC) and online platforms like *CodeHS.com* [21], block-based programming has become an integral part of introductory computer science education. Considering HOC alone, over one billion hours of block-based programming activity has been performed so far by over 50 million unique students worldwide [24, 35].

The societal need for enhancing K-12 computing education has led to a surge of interest in developing AI-driven systems for pedagogy of block-based programming [33, 26, 27, 34, 16]. Existing works have studied various aspects of intelligent support, including providing real-time next-step hints when a student is stuck solving a task [20, 36, 18, 17, 9], giving data-driven feedback about a student's misconceptions [31, 19, 28, 30, 35], and demonstrating a worked-out solution for a task when a student lacks the required programming concepts [37]. An underlying assumption when providing such intelligent support is that afterwards the student can practice *new similar tasks* to finally learn the missing concepts. However, this assumption is far from reality in existing systems—the programming tasks are typically hand-curated by experts/tutors, and the available set of tasks is limited. Consider HOC's *Classic Maze* challenge [23], which provides a progression of 20 tasks: Millions of students have attempted these tasks, yet when students fail to solve a task and receive assistance, they cannot practice similar tasks, hindering their ability to master the desired concepts. We seek to tackle this pedagogical challenge by developing techniques for synthesizing new programming tasks.

---

[*]Authors listed alphabetically; Correspondence to: Ahana Ghosh <gahana@mpi-sws.org>.

(a) Visual puzzle for $T^{in}$    (b) Solution code $C^{in}$    (c) Visual puzzle for $T^{out}$    (d) Solution code $C^{out}$

Figure 1: Illustration of our methodology for task *Maze 16* from the *Hour of Code: Classic Maze* challenge by *Code.org* [23]; the complete list of tasks with their specifications is in Fig. 6.

(a) Visual puzzle for $T^{in}$    (b) Solution code $C^{in}$    (c) Visual puzzle for $T^{out}$    (d) Solution code $C^{out}$

Figure 2: Illustration of our methodology for task *Diagonal* from the *Intro to Programming with Karel* course by *CodeHS.com* [22]; the complete list of tasks with their specifications is in Fig. 6.

We formalize the problem of synthesizing visual programming tasks of the kind found in popular learning platforms like *Code.org* (see Fig. 1) and *CodeHS.com* (see Fig. 2). As input, we are given a reference task $T^{in}$, specified as a visual puzzle, and its solution code $C^{in}$. Our goal is to synthesize a set $\{(T^{out}, C^{out})\}$ of new tasks along with their solution codes that are *conceptually similar* but *visually dissimilar* to the input. This is motivated by the need for practice tasks that on one hand exercise the same concepts, while looking fresh in order to maintain student engagement.

When tackling the problem of synthesizing new tasks with the above desirable properties, three key challenges emerge. First, we are generating problems in a *conceptual domain* with no *well-defined procedure* that students follow to solve a task—consequently, existing work on educational problem generation in procedural domains does not apply in our setting [3, 11]. Second, the mapping from the space of visual tasks to their solution codes is highly discontinuous; hence, template-based problem generation techniques [32, 25] that rely on directly mutating the input to generate new tasks is ineffective (see Section 5 where we use this approach as a baseline). Furthermore, such a direct task-mutation approach would require access to an automated solution synthesizer; however, state-of-the-art program synthesis techniques are not yet on par with experts and their minimal solutions [5, 8, 6]. Third, the space of possible tasks and their solutions is potentially unbounded, and thus, any problem generation technique that relies on exhaustive enumeration is intractable [32, 1, 2].

To overcome these challenges, we propose a novel methodology that operates by first mutating the solution code $C^{in}$ to obtain a set of codes $\{C^{out}\}$, and then performing symbolic execution over a code $C^{out}$ to obtain a visual puzzle $T^{out}$. Mutation is efficient by creating an abstract representation of $C^{in}$ along with appropriate constraints and querying an SMT solver [4]; any solution to this query is a mutated code $C^{out}$. During symbolic execution, we use Monte Carlo Tree Search (MCTS) to guide the search over the (unbounded) symbolic execution tree. We demonstrate the effectiveness of our methodology by performing an extensive empirical evaluation and user study on a set of reference tasks from the *Hour of code* challenge by *Code.org* and the *Intro to Programming with Karel* course by *CodeHS.com*. In summary, our main contributions are:

- We formalize the problem of synthesizing block-based visual programming tasks (Section 2).
- We present a novel approach for generating new visual tasks along with solution codes such that they are conceptually similar but visually dissimilar to a given reference task (Section 3).
- We demonstrate the effectiveness of our approach through an extensive empirical evaluation and user study on reference tasks from real-world programming platforms (Section 4 and Section 5).

## 2 Problem Formulation

**The space of tasks.** We define a task as a tuple $T := (T_{vis}, T_{store}, T_{size})$, where $T_{vis}$ denotes the visual puzzle, $T_{store}$ the available block types, and $T_{size}$ the maximum number of blocks allowed in the

solution code. For instance, considering the task $\mathtt{T} := \mathtt{T}^{\text{in}}$ in Fig. 1a, $\mathtt{T}_{\text{vis}}$ is illustrated in Fig. 1a, $\mathtt{T}_{\text{store}} = \{\mathtt{move}, \mathtt{turnL}, \mathtt{turnR}, \textsc{RepeatUntil}, \textsc{If}\}$, and $\mathtt{T}_{\text{size}} = 4$.

**The space of codes.** The programming environment has a domain-specific language (DSL), which defines the set of valid codes $\mathcal{C}$ and is shown in Fig. 4a. A code $\mathtt{C} \in \mathcal{C}$ is characterized by several properties, such as the set $\mathtt{C}_{\text{blocks}}$ of block types in $\mathtt{C}$, the number of blocks $\mathtt{C}_{\text{size}}$, the depth $\mathtt{C}_{\text{depth}}$ of the corresponding Abstract Syntax Tree (AST), and the nesting structure $\mathtt{C}_{\text{struct}}$ representing programming concepts exercised by $\mathtt{C}$. For instance, considering the code $\mathtt{C} := \mathtt{C}^{\text{in}}$ in Fig. 1b, $\mathtt{C}_{\text{blocks}} = \{\mathtt{move}, \mathtt{turnL}, \textsc{RepeatUntil}, \textsc{If}\}$, $\mathtt{C}_{\text{size}} = 4$, $\mathtt{C}_{\text{depth}} = 3$, and $\mathtt{C}_{\text{struct}} = \{\textsc{Run}\{\textsc{RepeatUntil}\{\textsc{If}\}\}\}$.

Below, we introduce two useful definitions relating the task and code space.

**Definition 1** (Solution code). *$\mathcal{C}$ is a solution code for $\mathtt{T}$ if the following holds: $\mathcal{C}$ successfully solves the visual puzzle $\mathtt{T}_{\text{vis}}$, $\mathcal{C}_{\text{blocks}} \subseteq \mathtt{T}_{\text{store}}$, and $\mathcal{C}_{\text{size}} \leq \mathtt{T}_{\text{size}}$. $\mathcal{C}_\mathtt{T}$ denotes the set of all solution codes for $\mathtt{T}$.*

**Definition 2** (Minimality of a task). *Given a solvable task $\mathtt{T}$ with $|\mathcal{C}_\mathtt{T}| \geq 1$ and a threshold $\delta \in \mathbb{N}$, the task is minimal if $\nexists \mathcal{C} \in \mathcal{C}_\mathtt{T}$ such that $\mathcal{C}_{\text{size}} < \mathtt{T}_{\text{size}} - \delta$.*

Next, we introduce two definitions formalizing the notion of conceptual similarity. Definition 3 formalizes conceptual similarity of a task $\mathtt{T}$ along with one solution code $\mathtt{C}$. Since a task can have multiple solution codes, Definition 4 provides a stricter notion of conceptual similarity of a task $\mathtt{T}$ for all its solution codes. These definitions are used in our objective of task synthesis in conditions (I) and (V) below.

**Definition 3** (Conceptual similarity of $(\mathtt{T}, \mathtt{C})$). *Given a reference $(\mathtt{T}^{\text{in}}, \mathtt{C}^{\text{in}})$ and a threshold $\delta \in \mathbb{N}$, a task $\mathtt{T}$ along with a solution code $\mathtt{C}$ is conceptually similar to $(\mathtt{T}^{\text{in}}, \mathtt{C}^{\text{in}})$ if the following holds: $\mathtt{T}_{\text{store}} = \mathtt{T}^{\text{in}}_{\text{store}}$, $|\mathtt{T}_{\text{size}} - \mathtt{T}^{\text{in}}_{\text{size}}| \leq \delta$, and $\mathtt{C}_{\text{struct}} = \mathtt{C}^{\text{in}}_{\text{struct}}$.*

**Definition 4** (Conceptual similarity of $(\mathtt{T}, \cdot)$). *Given a reference $(\mathtt{T}^{\text{in}}, \mathtt{C}^{\text{in}})$ and a threshold $\delta \in \mathbb{N}$, a task $\mathtt{T}$ is conceptually similar to $(\mathtt{T}^{\text{in}}, \mathtt{C}^{\text{in}})$ if the following holds: $\mathtt{T}_{\text{store}} = \mathtt{T}^{\text{in}}_{\text{store}}$, $|\mathtt{T}_{\text{size}} - \mathtt{T}^{\text{in}}_{\text{size}}| \leq \delta$, and $\forall \mathcal{C} \in \mathcal{C}_\mathtt{T}, \mathcal{C}_{\text{struct}} = \mathtt{C}^{\text{in}}_{\text{struct}}$.*

**Environment domain knowledge.** We now formalize our domain knowledge about the block-based environment to measure *visual dissimilarity* of two tasks, and capture some notion of *interestingness and quality* of a task. Given tasks $\mathtt{T}$ and $\mathtt{T}'$, we measure their visual dissimilarity by an environment-specific function $\mathcal{F}_{\text{diss}}(\mathtt{T}_{\text{vis}}, \mathtt{T}'_{\text{vis}}) \in [0, 1]$. Moreover, we measure generic quality of a task with function $\mathcal{F}_{\text{qual}}(\mathtt{T}_{\text{vis}}, \mathtt{C}) \in [0, 1]$. We provide specific instantiations of $\mathcal{F}_{\text{diss}}$ and $\mathcal{F}_{\text{qual}}$ in our evaluation.

**Objective of task synthesis.** Given a reference task $\mathtt{T}^{\text{in}}$ and a solution code $\mathtt{C}^{\text{in}} \in \mathcal{C}_{\mathtt{T}^{\text{in}}}$ as input, we seek to generate a set $\{(\mathtt{T}^{\text{out}}, \mathtt{C}^{\text{out}})\}$ of new tasks along with solution codes that are *conceptually similar* but *visually dissimilar* to the input. Formally, given parameters $(\delta_{\text{size}}, \delta_{\text{diss}}, \delta_{\text{qual}})$, our objective is to synthesize new tasks meeting the following conditions:

(I) $(\mathtt{T}^{\text{out}}, \mathtt{C}^{\text{out}})$ is conceptually similar to $(\mathtt{T}^{\text{in}}, \mathtt{C}^{\text{in}})$ with threshold $\delta_{\text{size}}$ in Definition 3.

(II) $\mathtt{T}^{\text{out}}$ is visually dissimilar to $\mathtt{T}^{\text{in}}$ with margin $\delta_{\text{diss}}$, i.e., $\mathcal{F}_{\text{diss}}(\mathtt{T}^{\text{in}}_{\text{vis}}, \mathtt{T}^{\text{out}}_{\text{vis}}) \geq \delta_{\text{diss}}$.

(III) $\mathtt{T}^{\text{out}}$ has a quality score above threshold $\delta_{\text{qual}}$, i.e., $\mathcal{F}_{\text{qual}}(\mathtt{T}^{\text{out}}_{\text{vis}}, \mathtt{C}^{\text{out}}) \geq \delta_{\text{qual}}$.

In addition, depending on the use case, it is desirable that the new tasks satisfy the following criteria:

(IV) $\mathtt{C}^{\text{out}}$ is different from the input solution code, i.e., $\mathtt{C}^{\text{out}} \neq \mathtt{C}^{\text{in}}$.

(V) $\mathtt{T}^{\text{out}}$ is conceptually similar to $(\mathtt{T}^{\text{in}}, \mathtt{C}^{\text{in}})$ with threshold $\delta_{\text{size}}$ in Definition 4.

(VI) $\mathtt{T}^{\text{out}}$ is minimal as per Definition 2 for a desired value of $\delta_{\text{mini}}$ (e.g., $\delta_{\text{mini}} = 0$ or $\delta_{\text{mini}} = 1$).

## 3 Our Task Synthesis Algorithm

We now present the pipeline of our algorithm (see Fig. 3), which takes as input a reference task $\mathtt{T}^{\text{in}}$ and its solution code $\mathtt{C}^{\text{in}}$, and generates a set $\{(\mathtt{T}^{\text{out}}, \mathtt{C}^{\text{out}})\}$ of new tasks with their solution codes. The goal is for this set to be conceptually similar to $(\mathtt{T}^{\text{in}}, \mathtt{C}^{\text{in}})$, but for new tasks $\{\mathtt{T}^{\text{out}}\}$ to

Figure 3: Stages in our task synthesis algorithm.

be visually dissimilar to $\mathtt{T}^{\text{in}}$. This is achieved by two main stages: (1) mutation of $\mathtt{C}^{\text{in}}$ to obtain a set $\{\mathtt{C}^{\text{out}}\}$, and (2) symbolic execution of each $\mathtt{C}^{\text{out}}$ to create a task $\mathtt{T}^{\text{out}}$. The first stage, presented in Section 3.1, converts $\mathtt{C}^{\text{in}}$ into an abstract representation restricted by a set of constraints (Fig. 3(a)), which must be satisfied by any generated $\mathtt{C}^{\text{out}}$ (Fig. 3(b)). The second stage, described in Section 3.2, applies symbolic execution on each code $\mathtt{C}^{\text{out}}$ to create a corresponding visual task $\mathtt{T}^{\text{out}}$ (Fig. 3(c)) while using Monte Carlo Tree Search (MCTS) to guide the search in the symbolic execution tree.

```
code C    := def RUN () DO y
rule y    := s | g | s; g
rule s    := a | s; s | IF (b) DO s | IF (b) DO s ELSE s
             | WHILE (b) DO s | REPEAT (x) DO s
rule g    := REPEATUNTIL (goal) DO s
action a  := move | turnL | turnR | putM | pickM
bool b    := pathA | noPathA | pathL | noPathL
             | pathR | noPathR | marker | noMarker
iter x    := 2 | 3 | 4 | 5 | 6 | 7 | 8 | 9 | 10
```

(a) Code DSL

```
sketch Q  := def RUN () DO Y
rule Y    := S | G | S; G
rule S    := A | S; S | IF (B) DO S | IF (B) DO S ELSE S
             | WHILE (B) DO S | REPEAT (X) DO S
rule G    := REPEATUNTIL (goal) DO S
Comments: A may be φ or take values of action a
          Ā denotes a sequence A₁,...,Aₙ
```

(b) Sketch DSL

**Input**: code C, sketch $Q \leftarrow \Omega(C)$, map $\omega(\cdot \mid C)$, $\delta_{\text{size}}$, $\delta_{\text{iter}}$

$(\Delta_0)$ Size of generated code may be at most $C_{\text{size}} + \delta_{\text{size}}$

$(\Delta_1)$ Edit action sequences $\text{ACTIONEDITS}(\{\bar{A} \in Q\}, \omega(\cdot \mid C))$

$(\Delta_2)$ For each $X \in Q : |X - \omega(X \mid C)| \le \delta_{\text{iter}}$

$(\Delta_3)$ Constraints induced by structure $\{\bar{A}_{\text{before}}; \text{REPEAT}\{\bar{A}\}; \bar{A}_{\text{after}}\}$

  i. $\bar{A}$ is not a suffix of $\bar{A}_{\text{before}}$

  ii. $\bar{A}$ is not a prefix of $\bar{A}_{\text{after}}$

$(\Delta_4)$ For each $B \in Q$ :

  i. $\omega(B \mid C) \in \{\text{pathA, noPathA}\}$
     $\Rightarrow B \in \{\text{pathA, noPathA}\}$

  ii. $\omega(B \mid C) \in \{\text{pathL, noPathL pathR, noPathR}\}$
     $\Rightarrow B \in \{\text{pathL, noPathL, pathR, noPathR}\}$

  iii. $\omega(B \mid C) \in \{\text{marker, noMarker}\}$
     $\Rightarrow B \in \{\text{marker, noMarker}\}$

$(\Delta_5)$ Constraints induced on $\bar{A}$ nested inside conditional $B$

$(\Delta_6)$ For each $\bar{A} \in Q$, constraints ensuring minimality of $\bar{A}$

(c) Types of Sketch Constraints

```
def RUN(){
  REPEATUNTIL(goal){
    move
    IF(pathLeft){
      turnLeft
    }
  }
}
```

(d) Code C^in

```
def RUN(){
  A₁¹, A₁² (Ā₁)
  REPEATUNTIL(goal){
    A₂¹, A₂², A₂³, A₂⁴, A₂⁵ (Ā₂)
    IF( B₁ ){
      A₃¹, A₃², A₃³, A₃⁴, A₃⁵ (Ā₃)
    }
  }
}
```

(e) Sketch Q^in

**Input**: $C^{\text{in}}$, $Q^{\text{in}}$, $\omega(\cdot \mid C^{\text{in}})$, $\delta_{\text{size}} = 2$

$(\Delta_0)$ Up to 2 new actions may be added in total to $\bar{A}_1, \bar{A}_2, \bar{A}_3$

$(\Delta_1)$ Edit action sequences $\text{ACTIONEDITS}(\{\bar{A}_1, \bar{A}_2, \bar{A}_3\}, \omega(\cdot \mid C^{\text{in}}))$

$(\Delta_4)$ $B_1 = \text{pathL} \lor B_1 = \text{pathR}$

$(\Delta_5)$ $\exists i \in [5]$ s.t. $\left(A_3^i = \text{turnL} \land (\forall j < i, A_3^j \notin \{\text{move, turnR}\})\right)$

$(\Delta_5)$ $\exists i \in [5]$ s.t. $\left(A_3^i = \text{turnR} \land (\forall j < i, A_3^j \notin \{\text{move, turnL}\})\right)$

$(\Delta_6)$ $\bar{A}_1, \bar{A}_2, \bar{A}_3$ are minimal

(f) Q^in-Constraints

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

 $\text{C}^{\text{out}}$ by task $\text{T}_{\text{vis}}^{\text{out}}$ and 0 otherwise; (ii) $\mathcal{F}_{\text{diss}}(\text{T}_{\text{vis}}^{\text{out}}, \text{T}_{\text{vis}}^{\text{in}}) \in [0, 1]$, which evaluates the dissimilarity of $\text{T}^{\text{out}}$ to $\text{T}^{\text{in}}$ (see Section 2); (iii) $\mathcal{F}_{\text{qual}}(\text{T}_{\text{vis}}^{\text{out}}, \text{C}^{\text{out}}) \in [0, 1]$, which evaluates the quality and validity of $\text{T}^{\text{out}}$; (iv) $\mathcal{F}_{\text{nocrash}}(\text{T}_{\text{vis}}^{\text{out}}, \text{C}^{\text{out}}) \in \{0, 1\}$, which evaluates to 0 in case the agent crashes into a wall and 1 otherwise; and (v) $\mathcal{F}_{\text{nocut}}(\text{T}_{\text{vis}}^{\text{out}}, \text{C}^{\text{out}}) \in \{0, 1\}$, which evaluates to 0 if there is a *shortcut sequence* of actions (a in Fig. 4a) smaller than $\text{C}_{\text{size}}^{\text{out}}$ that solves $\text{T}^{\text{out}}$ and 1 otherwise. $\mathcal{F}_{\text{qual}}$ and $\mathcal{F}_{\text{nocut}}$ also resolve the limitations of our mutation stage by eliminating codes and tasks that lead to undesirable agent behavior. We instantiate $\mathcal{F}_{\text{score}}$ in the next section.

## 4 Experimental Evaluation

In this section, we evaluate our task synthesis algorithm on HOC and Karel tasks. Our implementation is publicly available.[2] While we give an overview of key results here, a detailed description of our setup and additional experiments can be found in the supplementary material.

### 4.1 Reference Tasks and Specifications

**Reference tasks.** We use a set of ten reference tasks from HOC and Karel, shown in Fig. 6. The HOC tasks were selected from the *Hour of Code: Classic Maze* challenge by *Code.org* [23] and the Karel tasks from the *Intro to Programming with Karel* course by *CodeHS.com* [22]. The DSL of Fig. 4a is generic in that it includes both HOC and Karel codes, with the following differences: (i) construct WHILE, marker-related actions putM, pickM, and conditions noPathA, noPathL, noPathR, marker, noMarker are specific to Karel only; (ii) construct REPEATUNTIL and goal are specific to HOC only. Furthermore, the puzzles for HOC and Karel are of different styles (see Fig. 1 and Fig. 2). For all tasks, the grid size of the puzzles is fixed to $10 \times 10$ cells (grid-size parameter $n = 10$).

**Specification of scoring functions.** $\mathcal{F}_{\text{qual}}(\text{T}_{\text{vis}}^{\text{out}}, \text{C}^{\text{out}}) \in [0, 1]$ was approximated as the sum of the normalized counts of 'moves', 'turns', 'segments', and 'long-segments' in the grid; segments and long-segments are sequences of $\geq 3$ and $\geq 5$ move actions respectively. More precisely, for HOC tasks, we used the following function where features are computed by executing $\text{C}^{\text{out}}$ on $\text{T}_{\text{vis}}^{\text{out}}$:

$$\mathcal{F}_{\text{qual}}^{\text{HOC}}(\text{T}_{\text{vis}}^{\text{out}}, \text{C}^{\text{out}}) = \frac{1}{4}\Big(\frac{\#\text{moves}}{2n} + \frac{\#\text{turns}}{n} + \frac{\#\text{segments}}{n/2} + \frac{\#\text{long-segments}}{n/3}\Big).$$

Furthermore, in our implementation, $\mathcal{F}_{\text{qual}}(\cdot)$ value was set to 0 when $\mathcal{F}_{\text{nocrash}}(\cdot) = 0$. For Karel tasks, $\mathcal{F}_{\text{qual}}$ additionally included the normalized counts of putM and pickM, and is provided in the supplementary material. $\mathcal{F}_{\text{diss}}(\text{T}_{\text{vis}}^{\text{out}}, \text{T}_{\text{vis}}^{\text{in}}) \in [0, 1]$ was computed based on the dissimilarity of the agent's initial location/orientation w.r.t. $\text{T}_{\text{vis}}^{\text{in}}$, and the grid-cell level dissimilarity based on the Hamming distance between $\text{T}_{\text{vis}}^{\text{out}}$ and $\text{T}_{\text{vis}}^{\text{in}}$. More precisely, we used the following function:

$$\mathcal{F}_{\text{diss}}(\text{T}_{\text{vis}}^{\text{out}}, \text{T}_{\text{vis}}^{\text{in}}) = \frac{1}{3}\Big(\text{diss}(\text{loc} \mid \text{T}_{\text{vis}}^{\text{out}}, \text{T}_{\text{vis}}^{\text{in}}) + \text{diss}(\text{dir} \mid \text{T}_{\text{vis}}^{\text{out}}, \text{T}_{\text{vis}}^{\text{in}}) + \text{diss}(\text{grid-cells} \mid \text{T}_{\text{vis}}^{\text{out}}, \text{T}_{\text{vis}}^{\text{in}})\Big)$$

where $\text{diss}(\text{loc} \mid \text{T}_{\text{vis}}^{\text{out}}, \text{T}_{\text{vis}}^{\text{in}}) \in \{0, 1\}$, $\text{diss}(\text{dir} \mid \text{T}_{\text{vis}}^{\text{out}}, \text{T}_{\text{vis}}^{\text{in}}) \in \{0, 1\}$, and $\text{diss}(\text{grid-cells} \mid \text{T}_{\text{vis}}^{\text{out}}, \text{T}_{\text{vis}}^{\text{in}}) \in [0, 1]$ (after the Hamming distance is normalized with a factor of $\frac{2}{n^2}$).

| Task $T^{in}$ | Code Mutation | | | | Symbolic Execution | | | Fraction of $T^{out}$ with criteria | | |
|---|---|---|---|---|---|---|---|---|---|---|
| | $2{:}\#C^{out}_{\Delta=0}$ | $3{:}\#C^{out}_{\Delta=0,1}$ | $4{:}\#C^{out}_{\Delta=all}$ | $5{:}$Time | $6{:}\#C^{out}$ | $7{:}\#T^{out}$ | $8{:}$Time | $9{:}$(V) | $10{:}$(VI)$_{\delta_{mini}=1}$ | $11{:}$(VI)$_{\delta_{mini}=0}$ |
| H1 | $3,159$ | 112 | 64 | 0.6s | 28 | 272 | 68s | 1.00 | 1.00 | 1.00 |
| H2 | $8,991$ | 594 | 138 | 1.7s | 48 | 428 | 61s | 1.00 | 1.00 | 1.00 |
| H3 | $798,255$ | $13,122$ | 720 | 13.3s | 196 | $1,126$ | 60s | 0.90 | 0.98 | 0.90 |
| H4 | $5,913$ | 152 | 108 | 1.0s | 44 | 404 | 167s | 1.00 | 1.00 | 0.50 |
| H5 | $2,997$ | 294 | 66 | 0.8s | 46 | 444 | 348s | 0.98 | 0.59 | 0.27 |
| H6 | $1,728$ | 294 | 54 | 0.6s | 48 | 480 | 347s | 0.80 | 0.45 | 0.07 |
| K7 | $96,875$ | 150 | 122 | 1.3s | 122 | $1,196$ | 61s | 1.00 | 1.00 | 1.00 |
| K8 | $484,875$ | $4,506$ | 990 | 11.6s | 469 | $4,506$ | 63s | 1.00 | 1.00 | 1.00 |
| K9 | $8.595 \times 10^6$ | $60,768$ | 888 | 11.3s | 432 | $4,258$ | 185s | 0.92 | 0.92 | 0.88 |
| K10 | $132.625 \times 10^6$ | $19,328$ | $1,404$ | 17.1s | 532 | $5,032$ | 158s | 1.00 | 1.00 | 1.00 |

Figure 7: Results on HOC and Karel tasks; details are provided in Section 4.

Next, we define the evaluation function $\mathcal{F}_{score}(T^{out}, C^{out}, T^{in}, C^{in}) \in [0, 1]$ used by MCTS:

$$\mathcal{F}_{score}(T^{out}, C^{out}, T^{in}, C^{in}) = \underbrace{\mathbb{1}\Big(\

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

[4]The actual memory footprint of the MCTS procedure is small given that the symbolic tree is dynamically constructed during a run and the actual number of leaves explored is much lesser. In fact, the average runtime per output task (i.e., one MCTS run) as reported in Column 8 of Fig. 7 is achieved on a laptop machine with 2.8 GHz Quad-Core Intel Core i7 processor and 16 GB RAM.

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

# A  List of Appendices

In this section, we provide a brief description of the content provided in the appendices of the paper.

- Appendix B shows all the 10 reference tasks from Fig. 6. For each task, we also illustrate our methodology as was done in Fig. 1 and Fig. 2.
- Appendix C expands on the user study analysis. (Section 5)
- Appendix D provides additional details on the code mutation stage. (Section 3.1)
- Appendix E demonstrates how MCTS could guide the symbolic execution in generating more suitable tasks. (Section 3.2)
- Appendix F provides additional details and results about experiments. (Section 4)

# B  Illustration of Our Methodology for the HOC and Karel Dataset

In this section, we illustrate our methodology for all the 10 reference tasks from Fig. 6. For each reference task ($T^{in}$, $C^{in}$), we picked one output ($T^{out}$, $C^{out}$) from the entire set of generated outputs.

(a) Visual puzzle for $T^{in}$    (b) Solution code $C^{in}$    (c) Visual puzzle for $T^{out}$    (d) Solution code $C^{out}$

Figure 11: Task H1 – Illustration of our methodology.

(a) Visual puzzle for $T^{in}$    (b) Solution code $C^{in}$    (c) Visual puzzle for $T^{out}$    (d) Solution code $C^{out}$

Figure 12: Task H2 – Illustration of our methodology.

(a) Visual puzzle for $T^{in}$    (b) Solution code $C^{in}$    (c) Visual puzzle for $T^{out}$    (d) Solution code $C^{out}$

Figure 13: Task H3 – Illustration of our methodology.

(a) Visual puzzle for $T^{in}$    (b) Solution code $C^{in}$    (c) Visual puzzle for $T^{out}$    (d) Solution code $C^{out}$

Figure 14: Task H4 – Illustration of our methodology.

(a) Visual puzzle for $\text{T}^{\text{in}}$

```
def RUN(){
  REPEATUNTIL(goal){
    move
    IF(pathLeft){
      turnLeft
    }
  }
}
```

(b) Solution code $\text{C}^{\text{in}}$

(c) Visual puzzle for $\text{T}^{\text{out}}$

```
def RUN(){
  move
  turnLeft
  REPEATUNTIL(goal){
    move
    IF(pathRight){
      turnRight
    }
  }
}
```

(d) Solution code $\text{C}^{\text{out}}$

Figure 15: Task H5 – Illustration of our methodology (same as Fig. 1 and shown for completeness).

(a) Visual puzzle for $\text{T}^{\text{in}}$

```
def RUN(){
  REPEATUNTIL(goal){
    IF(pathAhead){
      move
    }
    ELSE{
      turnLeft
    }
  }
}
```

(b) Solution code $\text{C}^{\text{in}}$

(c) Visual puzzle for $\text{T}^{\text{out}}$

```
def RUN(){
  REPEATUNTIL(goal){
    IF(pathAhead){
      move
    }
    ELSE{
      turnRight
    }
  }
}
```

(d) Solution code $\text{C}^{\text{out}}$

Figure 16: Task H6 – Illustration of our methodology.

(a) Visual puzzle for $\text{T}^{\text{in}}$

```
def RUN(){
  move
  move
  pickMarker
  move
  move
}
```

(b) Solution code $\text{C}^{\text{in}}$

(c) Visual puzzle for $\text{T}^{\text{out}}$

```
def RUN(){
  putMarker
  turnLeft
  move
  move
  pickMarker
  move
  move
}
```

(d) Solution code $\text{C}^{\text{out}}$

Figure 17: Task K7 – Illustration of our methodology.

(a) Visual puzzle for $\text{T}^{\text{in}}$

```
def RUN(){
  REPEAT(4){
    putMarker
    move
    turnLeft
  }
}
```

(b) Solution code $\text{C}^{\text{in}}$

(c) Visual puzzle for $\text{T}^{\text{out}}$

```
def RUN(){
  REPEAT(4){
    move
    pickMarker
    move
    turnLeft
  }
}
```

(d) Solution code $\text{C}^{\text{out}}$

Figure 18: Task K8 – Illustration of our methodology.

```
def RUN(){
  REPEAT(8){
    IF(noMarker){
      putMarker
    }
    ELSE{
      pickMarker
    }
    move
  }
}
```

(b) Solution code $\text{C}^{\text{in}}$

(a) Visual puzzle for $\text{T}^{\text{in}}$

(c) Visual puzzle for $\text{T}^{\text{out}}$

```
def RUN(){
  turnRight
  REPEAT(7){
    IF(noMarker){
      putMarker
    }
    ELSE{
      pickMarker
    }
    move
  }
}
```

(d) Solution code $\text{C}^{\text{out}}$

Figure 19: Task K9 – Illustration of our methodology.

```
def RUN(){
  putMarker
  WHILE(pathAhead){
    move
    turnLeft
    move
    turnRight
    putMarker
  }
}
```

(b) Solution code $\text{C}^{\text{in}}$

(a) Visual puzzle for $\text{T}^{\text{in}}$

(c) Visual puzzle for $\text{T}^{\text{out}}$

```
def RUN(){
  putMarker
  WHILE(pathAhead){
    move
    move
    turnRight
    move
    turnLeft
    putMarker
  }
}
```

(d) Solution code $\text{C}^{\text{out}}$

Figure 20: Task K10 – Illustration of our methodology (same as Fig. 2 and shown for completeness).

# C  User Study: Additional Details and Results

In this section, we discuss additional details of our user study and expand on the key results provided in Section 5. For our user study, we used 4 reference tasks (H2, H4, H5, and H6). We developed an web app where participants, recruited through Amazon Mechanical Turk, were asked to solve tasks generated by four algorithms, SAME, TUTOR, MUTTASK, or SYNTASK (described in Section 5). Next, we describe the interface of our app and details of the questionnaire. The web app is publicly available (see Footnote 3).

(a) Login and welcome page

(b) Step 1: Coding task with solution

(c) Step 2: Solve new coding task

(d) Step 3: Survey

Figure 21: Interface of the app.

## C.1  App Interface and Questionnaire

Our online app was developed using the publicly available toolkit of Blockly Games [10] and provides an interface for a participant to practice Block-based programming tasks for HOC. Participants were familiarized with the tasks by a small tutorial given before they logged-in to the app. Each participant was randomly assigned a reference task, an algorithm (out of the four chosen for evaluation), and a particular task generated based on the chosen algorithm. These elements constituted a "practice session" for a participant. Each session consisted of three steps. In Step 1, the reference task along with its solution code was shown to the participant (Fig. 21b). In Step 2, the participant was asked to solve a new task (Fig. 21c). The new task was generated by one of the four algorithms: SAME, TUTOR, MUTTASK, or SYNTASK. To solve this new task, the participant was given 10 tries. If they successfully solved the task or ran out of tries, they were directed to Step 3 of the practice session, the survey (Fig. 21d). Here, they were presented with a question on the visual dissimilarity of the tasks from Step 1 and Step 2, which was to be answered on a 4-point Likert scale as used in [25]:

- Score 1: indicates that the tasks are visually exactly same

- Score 2: indicates that the tasks are visually similar

- Score 3: indicates that the tasks are visually slightly different

- Score 4: indicates that the tasks are visually very different

| (a) H- | (b) H2 | (c) H4 | (d) H5 | (e) H6 |

Figure 22: Distribution of participant responses on visual dissimilarity of tasks, rated on a 4-point Likert scale (1: exactly same, 2: almost same, 3: slightly different, 4: very different)

### C.2 Results on Visual Task Dissimilarity

The distribution of participant responses (on a 4-point scale) on the visual dissimilarity of the presented tasks is shown in Fig. 22; the aggregate statistics are provided in Fig. 9.

Comparing the dissimilarity of the generated tasks w.r.t. all the reference tasks H-, we found that the performance of SAME was worst (mean dissimilarity = 1.07) in comparison to MUTTASK (mean = 2.17), SYNTASK (mean = 2.63), and TUTOR (mean = 2.90). The poor performance of SAME, with the majority of scores being 1, is expected. Furthermore, we also see in Fig. 22a that MUTTASK has a greater percentage of lower scores of 1 and 2, and this is because the baseline directly mutates the task puzzle. Comparing the performance of SYNTASK to the baselines, we find that SYNTASK performed significantly better than MUTTASK ($\chi^2 = 38.81, p < e^{-7}$). However, its performance was worse than that of TUTOR and this difference was significant ($\chi^2 = 12.20, p = 0.0053$).

When analyzing the differences in the performance of SYNTASK w.r.t. TUTOR, we find that TUTOR performs better primarily because of the following two reasons: (i) some of the tasks generated by TUTOR have additional distracting paths / noise, and (ii) for simpler tasks without conditionals like H2, more powerful code mutations were used. Next, we performed additional analysis by limiting to two complex tasks with nested conditionals, H5 and H6. On these two tasks, the mean scores of the methods MUTTASK, SYNTASK, and TUTOR were 1.97, 2.94, and 3.03 respectively. Furthermore, we find that SYNTASK's performance continued to be significantly better than MUTTASK ($\chi^2 = 64.33, p < e^{-13}$). But, the difference in the performance of SYNTASK and TUTOR was not statistically significant ($\chi^2 = 2.68, p = 0.44$).

In general, the performance of our task synthesis algorithm can be further improved by allowing for more powerful code mutations in tasks without conditionals (such as H2) and by adding more variability in the output tasks by incorporating techniques discussed in Appendix F.4.

## D Code Mutation: Additional Details

This section describes the mutation stage of the task synthesis algorithm. In particular, we describe in detail the constraints applied on sketch $\mathbb{Q} = \Omega(\mathbb{C})$. Note that we denote an empty action as $\phi$. Our implementation of the code mutation stage, using the Z3 solver [7], is publicly available (see Footnote 2).

**Constraint ($\Delta_1$): ACTIONEDITS.** ACTIONEDITS returns the values that all action sequences can take, based on their values in the reference code, $\mathbb{C}^{\text{in}}$. Consider the action sequence $\overline{\mathbb{A}}$ in sketch $\mathbb{Q}$. The function $\omega(\overline{\mathbb{A}}|\,\mathbb{C}^{\text{in}})$ returns the value of $\overline{\mathbb{A}}$ in $\mathbb{C}^{\text{in}}$. The types of constraints returned by ACTIONEDITS are:

1. Local $\overline{\mathbb{A}}$ constraints. These constraints describe the values that one $\overline{\mathbb{A}}$ can take w.r.t. $\omega(\overline{\mathbb{A}}|\,\mathbb{C}^{\text{in}})$. It has the following rules:
   - `move` action $\in \omega(\overline{\mathbb{A}}|\,\mathbb{C}^{\text{in}})$, would imply that the corresponding action in $\overline{\mathbb{A}}$ must be `move`.
   - Set of `turnLeft`, `turnRight` actions $\in \omega(\overline{\mathbb{A}}|\,\mathbb{C}^{\text{in}})$, would imply that the corresponding actions in $\overline{\mathbb{A}}$ will either have all the 'turn' actions changed to `turnLeft`, or to `turnRight`, or remain the same, or be flipped, i.e. `turnLeft` $\rightarrow$ `turnRight` and `turnRight` $\rightarrow$ `turnLeft`.

- Set of `pickMarker`, `putMarker` actions $\in \omega(\overline{A}|\, C^{in})$, would imply that the corresponding actions in $\overline{A}$ will either have all the 'marker' actions changed to `pickMarker`, or to `putMarker`, or remain the same, or be flipped i.e. `pickMarker` $\rightarrow$ `putMarker` and `putMarker` $\rightarrow$ `pickMarker`.
- Additional actions (up to $\delta_{size}$) to $\overline{A}$ can either be appended before the existing actions (in $\omega(\overline{A}|\, C^{in})$) or after, but not both. In our experiments we set $\delta_{size} = 2$.

These constraints are listed under "Local $\overline{A}$ constraints" in Fig. 23h and Fig. 24h.

2. Global $\overline{A}$ constraints. These constraints apply on all the $\overline{A}$'s in sketch Q. They allow only one of all the $\overline{A}$'s to have additional actions added to them. These constraints are listed under "Global $\overline{A}$ constraints" in Fig. 23h and Fig. 24h.

**Constraint** $(\Delta_6)$**: Action sequence is minimal.** These constraints describe the sequences that invalidate minimality of code. The constraints ensure that certain sequences of actions do not occur in $\overline{A}$. The detailed list of sequences for the two example codes described, which invalidate $\overline{A}$ if they occur, are given in Fig. 23g and Fig. 24g.

### D.1 Details of Code Mutation for HOC

Here, we build on the generic DSL presented in Fig. 4 for HOC codes. We describe the HOC-DSL, Sketch DSL, and constraints for HOC codes (shown in Fig. 23a, Fig. 23b, and Fig. 23c, respectively). It is to be noted that the HOC DSL does not contain 'marker' based actions/conditionals: (action) `pickMarker`, (action) `putMarker`, (conditional) `marker`, (conditional) `noMarker`. It does not allow few more conditionals: `noPathA`, `noPathL` and `noPathR`. It also does not contain the WHILE construct. We consider a concrete example (continued from the example presented in Fig. 4d), in Fig. 23d, illustrate its sketch in Fig. 23e, and its constraints in Fig. 23f. We elaborate its $\overline{A}$ minimality constraints and ACTIONEDITS in Fig. 23g and Fig. 23h, respectively.

**A brief description of elimination sequences to ensure $\overline{A}$-minimality.** We identify four sequences in HOC codes, which must be removed to ensure code minimality. These are: `turnLeft`, `turnRight` or `turnRight`, `turnLeft`, which do not lead to any change in the output; `turnLeft`, `turnLeft`, `turnLeft`, which can be replaced by a single `turnRight`; and finally `turnRight`, `turnRight`, `turnRight`, which can be replaced by a single `turnLeft`. It is to be noted that we also eliminate variants of the above sequences, that contain $\phi$, but effectively reduce to these four sequences only, given that $\phi$ denotes an empty action.

### D.2 Details of Code Mutation for Karel

Here, we present a modified/detailed form of Fig. 4 for Karel codes in particular. We describe the Karel-DSL, Sketch DSL, and constraints for Karel codes (shown in Fig. 24a, Fig. 24b, and Fig. 24c, respectively). It is to be noted that Karel-DSL does not contain the REPEATUNTIL construct. We consider a concrete example (same as the Karel solution code presented in Fig. 2b), in Fig. 24d, illustrate its sketch in Fig. 24e, and its constraints in Fig. 24f. We elaborate its $\overline{A}$ minimality constraints and ACTIONEDITS in Fig. 24g and Fig. 24h, respectively.

**A brief description of elimination sequences to ensure $\overline{A}$-minimality.** In addition to the four sequences considered in HOC codes, we identify sixteen more sequences in Karel codes, which must be removed to ensure code minimality. These are: `pickMarker`, `putMarker` or `putMarker`, `pickMarker` or `pickMarker`, `pickMarker` or `putMarker`, `putMarker`, which do not lead to any change in the output or leads to a crash in the Karel grid (only one marker is allowed per grid-cell); `turnLeft`, `pickMarker`, `turnRight` or `turnRight`, `pickMarker`, `turnLeft` or `turnLeft`, `putMarker`, `turnRight` or `turnRight`, `putMarker`, `turnLeft`, which bring about the same output without the 'turn' actions; `pickMarker`, `turnLeft`, `putMarker` or `putMarker`, `turnLeft`, `pickMarker` or `pickMarker`, `turnRight`, `putMarker` or `putMarker`, `turnRight`, `pickMarker`, which bring about the same output without the 'marker' actions; and finally `pickMarker`, `turnLeft`, `pickMarker` or `pickMarker`, `turnRight`, `pickMarker` or `putMarker`, `turnLeft`, `putMarker` or `putMarker`, `turnRight`, `putMarker`, which leads to a crash in the Karel grid (only one marker is allowed per grid-cell). It is to be noted that we also eliminate variants of the above sequences, that contain $\phi$, but effectively reduce to these basic sequences only, given that $\phi$ denotes an empty action.

**(a) Code DSL – HOC**

```
code C    := def Run () do y
rule y    := s | g | s; g
rule s    := a | s; s | If (b) do s | If (b) do s Else s
             | Repeat (x) do s
rule g    := RepeatUntil (goal) do s
action a  := move | turnL | turnR
bool b    := pathA | pathL | pathR
iter x    := 2 | 3 | 4 | 5 | 6 | 7 | 8 | 9 | 10
```

**(b) Sketch DSL – HOC**

```
sketch Q  := def Run () do Y
rule Y    := S | G | S; G
rule S    := A | S; S | If (B) do S | If (B) do S Else S
             | Repeat (X) do S
rule G    := RepeatUntil (goal) do S
```
Comments: A may be $\phi$ or take values of action a
$\overline{A}$ denotes a sequence $A_1, \ldots, A_n$

**(c) Types of Sketch Constraints – HOC**

**Input**: code C, sketch $Q \leftarrow \Omega(C)$, map $\omega(\cdot \mid C)$, $\delta_{size}$, $\delta_{iter}$

$(\Delta_0)$ Size of generated code may be at most $C_{size} + \delta_{size}$

$(\Delta_1)$ Edit action sequences $\text{ActionEdits}(\{\overline{A} \in Q\}, \omega(\cdot \mid C))$

$(\Delta_2)$ For each $X \in Q : |X - \omega(X \mid C)| \leq \delta_{iter}$

$(\Delta_3)$ Constraints induced by structure $\{\overline{A}_{before}; \text{Repeat } \{\overline{A}\}; \overline{A}_{after}\}$

  i. $\overline{A}$ is not a suffix of $\overline{A}_{before}$

  ii. $\overline{A}$ is not a prefix of $\overline{A}_{after}$

$(\Delta_4)$ For each $B \in Q$ :

  i. $\omega(B \mid C) \in \{pathA\}$
      $\Rightarrow B \in \{pathA\}$

  ii. $\omega(B \mid C) \in \{pathL, pathR\}$
      $\Rightarrow B \in \{pathL, pathR\}$

$(\Delta_5)$ Constraints induced on $\overline{A}$ nested inside conditional B

$(\Delta_6)$ For each $\overline{A} \in Q$, constraints ensuring minimality of $\overline{A}$

**(d) Code $C^{in}$**

```
def Run(){
  RepeatUntil(goal){
    move
    If(pathLeft){
      turnLeft
    }
  }
}
```

**(e) Sketch $Q^{in}$**

```
def Run(){
  A_1^1, A_1^2 (Ā_1)
  RepeatUntil(goal){
    A_2^1, A_2^2, A_2^3, A_2^4, A_2^5 (Ā_2)
    If( B_1 ){
      A_3^1, A_3^2, A_3^3, A_3^4, A_3^5 (Ā_3)
    }
  }
}
```

**(f) $Q^{in}$-Constraints for HOC**

**Input**: $C^{in}$, $Q^{in}$, $\omega(\cdot \mid C^{in})$, $\delta_{size} = 2$

$(\Delta_0)$ Up to 2 new actions may be added in total to $\overline{A}_1, \overline{A}_2, \overline{A}_3$

$(\Delta_1)$ Edit action sequences $\text{ActionEdits}(\{\overline{A}_1, \overline{A}_2, \overline{A}_3\}, \omega(\cdot \mid C^{in}))$

$(\Delta_4)$ $B_1 = pathL \lor B_1 = pathR$

$(\Delta_5)$ $(B_1 = pathL) \Rightarrow$
$\left( \exists i \in [5] \text{ s.t } \left(A_3^i = \text{turnL} \ \land \ ( \forall j < i, A_3^j \notin \{\text{move}, \text{turnR}\}) \right) \right)$

$(\Delta_5)$ $(B_1 = pathR) \Rightarrow$
$\left( \exists i \in [5] \text{ s.t } \left(A_3^i = \text{turnR} \ \land \ ( \forall j < i, A_3^j \notin \{\text{move}, \text{turnL}\}) \right) \right)$

$(\Delta_6)$ $\overline{A}_1, \overline{A}_2, \overline{A}_3$ are minimal

**(g) $Q^{in}$-Constraints: $(\Delta_6)$ $\overline{A}_1, \overline{A}_2, \overline{A}_3$ are minimal**

Set of elimination sequences $\mathcal{E} := \{(L, R), (R, L), (L, L, L), (R, R, R)\}$

(i) Apply each elimination sequence from $\mathcal{E}$ to $\overline{A}_1$ as a set of constraints.

(ii) Apply each elimination sequence from $\mathcal{E}$ to $\overline{A}_2$ as a set of constraints.

(iii) Apply each elimination sequence from $\mathcal{E}$ to $\overline{A}_3$ as a set of constraints.

**(h) $Q^{in}$-Constraints: $(\Delta_1)$ $\text{ActionEdits}(\{\overline{A}_1, \overline{A}_2, \overline{A}_3\}, \omega(\cdot \mid C^{in}))$**

Shorthand notation for actions: $\text{move} \to M$, $\text{turnL} \to L$, $\text{turnR} \to R$, $\phi \to$ empty-action.

Local $\overline{A}$ constraints that define the values that each action sequence can take:

(i) $(A_1^1 = M \lor A_1^1 = L \lor A_1^1 = R \lor A_1^1 = \phi) \land (A_1^2 = M \lor A_1^2 = L \lor A_1^2 = R \lor A_1^2 = \phi)$

(ii) $(A_2^1 = M \lor A_2^1 = L \lor A_2^1 = R \lor A_2^1 = \phi) \land (A_2^2 = M \lor A_2^2 = L \lor A_2^2 = R \lor A_2^2 = \phi) \ (A_2^3 = M) \land (A_2^4 = M \lor A_2^4 = L \lor A_2^4 = R \lor A_2^4 = \phi) \land$
$(A_2^5 = M \lor A_2^5 = L \lor A_2^5 = R \lor A_2^5 = \phi)$

(iii) $(A_2^1 \neq \phi \lor A_2^2 \neq \phi) \Rightarrow (A_2^4 = \phi \land A_2^5 = \phi)$

(iv) $(A_2^4 \neq \phi \lor A_2^5 \neq \phi) \Rightarrow (A_2^1 = \phi \land A_2^2 = \phi)$

(v) $(A_3^1 = M \lor A_3^1 = L \lor A_3^1 = R \lor A_3^1 = \phi) \land (A_3^2 = M \lor A_3^2 = L \lor A_3^2 = R \lor A_3^2 = \phi) \ (A_3^3 = L \lor A_3^3 = R) \land$
$(A_3^4 = M \lor A_3^4 = L \lor A_3^4 = R \lor A_3^4 = \phi) \land (A_3^5 = M \lor A_3^5 = L \lor A_3^5 = R \lor A_3^5 = \phi)$

(vi) $(A_3^1 \neq \phi \lor A_3^2 \neq \phi) \Rightarrow (A_3^4 = \phi \land A_3^4 = \phi)$

(vii) $(A_3^4 \neq \phi \lor A_3^5 \neq \phi) \Rightarrow (A_3^1 = \phi \land A_3^2 = \phi)$

Global $\overline{A}$ constraints that allow actions to be added to either of $\overline{A}_1, \overline{A}_2, \overline{A}_3$:

(i) $(A_1^1 \neq \phi \lor A_1^2 \neq \phi) \Rightarrow (A_2^1 = \phi \land A_2^2 = \phi \land A_2^4 = \phi \land A_2^5 = \phi \land A_3^1 = \phi \land A_3^2 = \phi \land A_3^4 = \phi \land A_3^5 = \phi)$

(ii) $(A_2^1 \neq \phi \lor A_2^2 \neq \phi) \Rightarrow (A_1^1 = \phi \land A_1^2 = \phi \land A_3^1 = \phi \land A_3^2 = \phi \land A_3^4 = \phi \land A_3^5 = \phi)$
$(A_2^4 \neq \phi \lor A_2^5 \neq \phi) \Rightarrow (A_1^1 = \phi \land A_1^2 = \phi \land A_3^1 = \phi \land A_3^2 = \phi \land A_3^4 = \phi \land A_3^5 = \phi)$

(iii) $(A_3^1 \neq \phi \lor A_3^2 \neq \phi) \Rightarrow (A_1^1 = \phi \land A_1^2 = \phi \land A_2^1 = \phi \land A_2^2 = \phi \land A_2^4 = \phi \land A_2^5 = \phi \land)$
$(A_3^4 \neq \phi \lor A_3^5 \neq \phi) \Rightarrow (A_1^1 = \phi \land A_1^2 = \phi \land A_2^1 = \phi \land A_2^2 = \phi \land A_2^4 = \phi \land A_2^5 = \phi)$

Figure 23: Illustration of Code Mutation for HOC on the solution code for task *Maze 16* from the *Hour of Code: Classic Maze* challenge by *Code.org* [23]; We build on Fig. 4 here.

| | |
|---|---|
| code C | := def RUN () DO s |
| rule s | := a \| s; s \| IF (b) DO s \| IF (b) DO s ELSE s |
| | \| WHILE (b) DO s \| REPEAT (x) DO s |
| action a | := move \| turnL \| turnR \| putM \| pickM |
| bool b | := pathA \| noPathA \| pathL \| noPathL |
| | \| pathR \| noPathR \| marker \| noMarker |
| iter x | := 2 \| 3 \| 4 \| 5 \| 6 \| 7 \| 8 \| 9 \| 10 |

(a) Code DSL – Karel

**Input**: code C, sketch $Q \leftarrow \Omega(C)$, map $\omega(\cdot \mid C)$, $\delta_{\text{size}}$, $\delta_{\text{iter}}$

($\Delta_0$) Size of generated code may be at most $C_{\text{size}} + \delta_{\text{size}}$

($\Delta_1$) Edit action sequences ACTIONEDITS($\{\bar{A} \in Q\}, \omega(\cdot \mid C)$)

($\Delta_2$) For each $X \in Q$ : $|X - \omega(X \mid C)| \leq \delta_{\text{iter}}$

($\Delta_3$) Constraints induced by structure $\{\bar{A}_{\text{before}}; \text{REPEAT} \{\bar{A}\}; \bar{A}_{\text{after}}\}$

  i. $\bar{A}$ is not a suffix of $\bar{A}_{\text{before}}$

  ii. $\bar{A}$ is not a prefix of $\bar{A}_{\text{after}}$

($\Delta_4$) For each $B \in Q$ :

  i. $\omega(B \mid C) \in \{\text{pathA, noPathA}\}$
     $\Rightarrow B \in \{\text{pathA, noPathA}\}$

  ii. $\omega(B \mid C) \in \{\text{pathL, noPathL pathR, noPathR}\}$
     $\Rightarrow B \in \{\text{pathL, noPathL, pathR, noPathR}\}$

  iii. $\omega(B \mid C) \in \{\text{marker, noMarker}\}$
     $\Rightarrow B \in \{\text{marker, noMarker}\}$

($\Delta_5$) Constraints induced on $\bar{A}$ nested inside conditional B

($\Delta_6$) For each $\bar{A} \in Q$, constraints ensuring minimality of $\bar{A}$

| | |
|---|---|
| sketch Q | := def RUN () DO Y |
| rule Y | := S |
| rule S | := A \| S; S \| IF (B) DO S \| IF (B) DO S ELSE S |
| | \| WHILE (B) DO S \| REPEAT (X) DO S |

Comments: A may be $\phi$ or take values of action a
        $\bar{A}$ denotes a sequence $A_1, \ldots, A_n$

(b) Sketch DSL – Karel

(c) Types of Sketch Constraints – Karel

```
def RUN(){
  putMarker
  WHILE(pathAhead){
    move
    turnLeft
    move
    turnRight
    putMarker
  }
}
```

(d) Code $C^{\text{in}}$

```
def RUN(){
  A₁¹, A₁², A₁³, A₁⁴, A₁⁵, (Ā₁)
  WHILE( B₁){
    A₂¹, A₂²,
    A₂³, A₂⁴, A₂⁵,
    A₂⁶, A₂⁷, A₂⁸,
    A₂⁹, A₂¹⁰, A₂¹¹,   (Ā₂)
    A₂¹², A₂¹³
  }
  A₃¹, A₃² (Ā₃)
}
```

(e) Sketch $Q^{\text{in}}$

**Input**: $C^{\text{in}}$, $Q^{\text{in}}$, $\omega(\cdot \mid C^{\text{in}})$, $\delta_{\text{size}} = 2$

($\Delta_0$) Up to 2 new actions may be added in total to $\bar{A}_1, \bar{A}_2, \bar{A}_3$

($\Delta_1$) Edit action sequences ACTIONEDITS($\{\bar{A}_1, \bar{A}_2, \bar{A}_3\}, \omega(\cdot \mid C^{\text{in}})$)

($\Delta_4$) $B_1 = \text{pathA} \lor B_1 = \text{noPathA}$

($\Delta_5$) $(B_1 = \text{pathA}) \Rightarrow$
$\left( \exists i \in [13] \text{ s.t } \left( A_2^i = \text{move} \land (\forall j < i, A_2^j \notin \{\text{turnL, turnR}\}) \right) \right)$

($\Delta_5$) $(B_1 = \text{noPathA}) \Rightarrow$
$\left( \exists i \in [13] \text{ s.t } \left( A_2^i = \text{move} \Rightarrow (\exists j < i, A_2^j \in \{\text{turnL, turnR}\}) \right) \right)$

($\Delta_6$) $\bar{A}_1, \bar{A}_2, \bar{A}_3$ are minimal

(f) $Q^{\text{in}}$-Constraints for Karel

Set of elimination sequences $\mathcal{E} := \{(\text{L, R}), (\text{R, L}), (\text{L, L, L}), (\text{R, R, R}), (\text{piM, puM}), (\text{puM, piM}), (\text{piM, piM}), (\text{puM, puM}), (\text{L, piM, R}),$
$(\text{R, piM, L}), (\text{L, puM, R}), (\text{R, puM, L}), (\text{piM, L, puM}), (\text{puM, L, piM}), (\text{piM, R, puM}), (\text{puM, R, piM}),$
$(\text{piM, L, piM}), (\text{piM, R, piM}), (\text{puM, L, puM}), (\text{puM, R, puM})\}$

 (i) Apply each elimination sequence from $\mathcal{E}$ to $\bar{A}_1$ as a set of constraints.
 (ii) Apply each elimination sequence from $\mathcal{E}$ to $\bar{A}_2$ as a set of constraints.
 (iii) Apply each elimination sequence from $\mathcal{E}$ to $\bar{A}_3$ as a set of constraints.

(g) $Q^{\text{in}}$-Constraints: ($\Delta_6$) $\bar{A}_1, \bar{A}_2, \bar{A}_3$ are minimal

Figure 24: **(a)–(g)**: Illustration of Code Mutation for Karel task *Diagonal* from the *Intro to Programming with Karel* course by *CodeHS.com* [22]; We present the Karel variant of Fig. 4 here.

Local $\overline{\texttt{A}}$ constraints that define the values that each action sequence can take:

(i) $\quad (\texttt{A}_1^1 = \texttt{M} \vee \texttt{A}_1^1 = \texttt{L} \vee \texttt{A}_1^1 = \texttt{R} \vee \texttt{A}_1^1 = \texttt{piM} \vee \texttt{A}_1^1 = \texttt{puM} \vee \texttt{A}_1^1 = \phi)$

$\quad \wedge (\texttt{A}_1^2 = \texttt{M} \vee \texttt{A}_1^2 = \texttt{L} \vee \texttt{A}_1^2 = \texttt{R} \vee \texttt{A}_1^2 = \texttt{piM} \vee \texttt{A}_1^2 = \texttt{puM} \vee \texttt{A}_1^2 = \phi)$

$\quad \wedge (\texttt{A}_1^3 = \texttt{piM} \vee \texttt{A}_1^3 = \texttt{puM})$

$\quad \wedge (\texttt{A}_1^4 = \texttt{M} \vee \texttt{A}_1^4 = \texttt{L} \vee \texttt{A}_1^4 = \texttt{R} \vee \texttt{A}_1^4 = \texttt{piM} \vee \texttt{A}_1^4 = \texttt{puM} \vee \texttt{A}_1^4 = \phi)$

$\quad \wedge (\texttt{A}_1^5 = \texttt{M} \vee \texttt{A}_1^5 = \texttt{L} \vee \texttt{A}_1^5 = \texttt{R} \vee \texttt{A}_1^5 = \texttt{piM} \vee \texttt{A}_1^5 = \texttt{puM} \vee \texttt{A}_1^5 = \phi)$

(ii) $\wedge (\texttt{A}_1^1 \neq \phi \vee \texttt{A}_1^2 \neq \phi) \Rightarrow (\texttt{A}_1^4 = \phi \wedge \texttt{A}_1^5 = \phi) \wedge$

$\quad \wedge (\texttt{A}_1^4 \neq \phi \vee \texttt{A}_1^5 \neq \phi) \Rightarrow (\texttt{A}_1^1 = \phi \wedge \texttt{A}_1^2 = \phi)$

(iii) $\quad (\texttt{A}_2^1 = \texttt{M} \vee \texttt{A}_2^1 = \texttt{L} \vee \texttt{A}_2^1 = \texttt{R} \vee \texttt{A}_2^1 = \texttt{piM} \vee \texttt{A}_2^1 = \texttt{puM} \vee \texttt{A}_2^1 = \phi)$

$\quad \wedge (\texttt{A}_2^2 = \texttt{M} \vee \texttt{A}_2^2 = \texttt{L} \vee \texttt{A}_2^2 = \texttt{R} \vee \texttt{A}_2^2 = \texttt{piM} \vee \texttt{A}_2^2 = \texttt{puM} \vee \texttt{A}_2^2 = \phi)$

$\quad \wedge (\texttt{A}_2^3 = \texttt{M})$

$\quad \wedge (\texttt{A}_2^4 = \texttt{M} \vee \texttt{A}_2^4 = \texttt{L} \vee \texttt{A}_2^4 = \texttt{R} \vee \texttt{A}_2^4 = \texttt{piM} \vee \texttt{A}_2^4 = \texttt{puM} \vee \texttt{A}_2^4 = \phi)$

$\quad \wedge \big((\texttt{A}_2^5 = \texttt{L} \wedge \texttt{A}_2^9 = \texttt{L}) \vee (\texttt{A}_2^5 = \texttt{R} \wedge \texttt{A}_2^9 = \texttt{R}) \vee (\texttt{A}_2^5 = \texttt{L} \wedge \texttt{A}_2^9 = \texttt{R}) \vee (\texttt{A}_2^5 = \texttt{R} \wedge \texttt{A}_2^9 = \texttt{L})\big)$

$\quad \wedge (\texttt{A}_2^6 = \texttt{M} \vee \texttt{A}_2^6 = \texttt{L} \vee \texttt{A}_2^6 = \texttt{R} \vee \texttt{A}_2^6 = \texttt{piM} \vee \texttt{A}_2^6 = \texttt{puM} \vee \texttt{A}_2^6 = \phi)$

$\quad \wedge (\texttt{A}_2^7 = \texttt{M})$

$\quad \wedge (\texttt{A}_2^8 = \texttt{M} \vee \texttt{A}_2^8 = \texttt{L} \vee \texttt{A}_2^8 = \texttt{R} \vee \texttt{A}_2^8 = \texttt{piM} \vee \texttt{A}_2^8 = \texttt{puM} \vee \texttt{A}_2^8 = \phi)$

$\quad \wedge (\texttt{A}_2^{10} = \texttt{M} \vee \texttt{A}_2^{10} = \texttt{L} \vee \texttt{A}_2^{10} = \texttt{R} \vee \texttt{A}_2^{10} = \texttt{piM} \vee \texttt{A}_2^{10} = \texttt{puM} \vee \texttt{A}_2^{10} = \phi)$

$\quad \wedge (\texttt{A}_2^{11} = \texttt{piM} \vee \texttt{A}_2^{11} = \texttt{puM})$

$\quad \wedge (\texttt{A}_2^{12} = \texttt{M} \vee \texttt{A}_2^{12} = \texttt{L} \vee \texttt{A}_2^{12} = \texttt{R} \vee \texttt{A}_2^{12} = \texttt{piM} \vee \texttt{A}_2^{12} = \texttt{puM} \vee \texttt{A}_2^{12} = \phi)$

$\quad \wedge (\texttt{A}_2^{13} = \texttt{M} \vee \texttt{A}_2^{13} = \texttt{L} \vee \texttt{A}_2^{13} = \texttt{R} \vee \texttt{A}_2^{13} = \texttt{piM} \vee \texttt{A}_2^{13} = \texttt{puM} \vee \texttt{A}_2^{13} = \phi)$

(iv) $\quad \big((\texttt{A}_2^1 \neq \phi \vee \texttt{A}_2^2 \neq \phi) \Rightarrow (\texttt{A}_2^4 = \phi \wedge \texttt{A}_2^6 = \phi \wedge \texttt{A}_2^8 = \phi \wedge \texttt{A}_2^{10} = \phi \wedge \texttt{A}_2^{12} = \phi \wedge \texttt{A}_2^{13} = \phi)\big)$

$\quad \wedge \big((\texttt{A}_2^4 \neq \phi) \Rightarrow (\texttt{A}_2^1 = \phi \wedge \texttt{A}_2^2 = \phi \wedge \texttt{A}_2^6 = \phi \wedge \texttt{A}_2^8 = \phi \wedge \texttt{A}_2^{10} = \phi \wedge \texttt{A}_2^{12} = \phi \wedge \texttt{A}_2^{13} = \phi)\big)$

$\quad \wedge \big((\texttt{A}_2^6 \neq \phi) \Rightarrow (\texttt{A}_2^1 = \phi \wedge \texttt{A}_2^2 = \phi \wedge \texttt{A}_2^4 = \phi \wedge \texttt{A}_2^8 = \phi \wedge \texttt{A}_2^{10} = \phi \wedge \texttt{A}_2^{12} = \phi \wedge \texttt{A}_2^{13} = \phi)\big)$

$\quad \wedge \big((\texttt{A}_2^8 \neq \phi) \Rightarrow (\texttt{A}_2^1 = \phi \wedge \texttt{A}_2^2 = \phi \wedge \texttt{A}_2^4 = \phi \wedge \texttt{A}_2^6 = \phi \wedge \texttt{A}_2^{10} = \phi \wedge \texttt{A}_2^{12} = \phi \wedge \texttt{A}_2^{13} = \phi)\big)$

$\quad \wedge \big((\texttt{A}_2^{10} \neq \phi) \Rightarrow (\texttt{A}_2^1 = \phi \wedge \texttt{A}_2^2 = \phi \wedge \texttt{A}_2^4 = \phi \wedge \texttt{A}_2^6 = \phi \wedge \texttt{A}_2^8 = \phi \wedge \texttt{A}_2^{12} = \phi \wedge \texttt{A}_2^{13} = \phi)\big)$

$\quad \wedge \big((\texttt{A}_2^{12} \neq \phi \vee \texttt{A}_2^{13} \neq \phi) \Rightarrow (\texttt{A}_2^1 = \phi \wedge \texttt{A}_2^2 = \phi \wedge \texttt{A}_2^4 = \phi \wedge \texttt{A}_2^6 = \phi \wedge \texttt{A}_2^8 = \phi \wedge \texttt{A}_2^{10} = \phi)\big)$

(v) $\quad (\texttt{A}_3^1 = \texttt{M} \vee \texttt{A}_3^1 = \texttt{L} \vee \texttt{A}_3^1 = \texttt{R} \vee \texttt{A}_3^1 = \texttt{piM} \vee \texttt{A}_3^1 = \texttt{puM} \vee \texttt{A}_3^1 = \phi)$

$\quad \wedge (\texttt{A}_3^2 = \texttt{M} \vee \texttt{A}_3^2 = \texttt{L} \vee \texttt{A}_3^2 = \texttt{R} \vee \texttt{A}_3^2 = \texttt{piM} \vee \texttt{A}_3^2 = \texttt{puM} \vee \texttt{A}_3^2 = \phi)$

Global $\overline{\texttt{A}}$ constraints that allow actions to be added to either of $\overline{\texttt{A}}_1, \overline{\texttt{A}}_2, \overline{\texttt{A}}_3$:

(i) $(\texttt{A}_1^1 \neq \phi \vee \texttt{A}_1^2 \neq \phi \vee \texttt{A}_1^3 \neq \phi \vee \texttt{A}_1^4 \neq \phi)$

$\quad \Rightarrow (\texttt{A}_2^1 = \phi \wedge \texttt{A}_2^2 = \phi \wedge \texttt{A}_2^4 = \phi \wedge \texttt{A}_2^6 = \phi \wedge \texttt{A}_2^8 = \phi \wedge \texttt{A}_2^{10} = \phi \wedge \texttt{A}_2^{12} = \phi \wedge \texttt{A}_2^{13} = \phi \wedge \texttt{A}_3^1 = \phi \wedge \texttt{A}_3^2 = \phi)$

(ii) $(\texttt{A}_2^1 \neq \phi \vee \texttt{A}_2^2 \neq \phi \vee \texttt{A}_2^4 \neq \phi \wedge \texttt{A}_2^6 \neq \phi \wedge \texttt{A}_2^8 \neq \phi \wedge \texttt{A}_2^{10} \neq \phi \wedge \texttt{A}_2^{12} \neq \phi \wedge \texttt{A}_2^{13} \neq \phi)$

$\quad \Rightarrow (\texttt{A}_1^1 = \phi \wedge \texttt{A}_1^2 = \phi \wedge \texttt{A}_1^4 = \phi \wedge \texttt{A}_1^5 = \phi \wedge \texttt{A}_3^1 = \phi \wedge \texttt{A}_3^2 = \phi)$

(iii) $(\texttt{A}_3^1 \neq \phi \vee \texttt{A}_3^2 \neq \phi)$

$\quad \Rightarrow (\texttt{A}_1^1 = \phi \wedge \texttt{A}_1^2 = \phi \wedge \texttt{A}_1^4 = \phi \wedge \texttt{A}_1^5 = \phi \wedge \texttt{A}_2^1 = \phi \wedge \texttt{A}_2^2 = \phi \wedge \texttt{A}_2^4 = \phi \wedge \texttt{A}_2^6 = \phi \wedge \texttt{A}_2^8 = \phi \wedge \texttt{A}_2^{10} = \phi \wedge \texttt{A}_2^{12} = \phi \wedge \texttt{A}_2^{13} = \phi)$

(h) $\texttt{Q}^{\text{in}}$-Constraints: $(\Delta_1)$ ACTIONEDITS$(\{\overline{\texttt{A}}_1, \overline{\texttt{A}}_2, \overline{\texttt{A}}_3\}, \omega(\cdot \,|\, \texttt{C}^{\text{in}}))$

Figure 24: (**h**): Illustration of Code Mutation for Karel task *Diagonal* from the *Intro to Programming with Karel* course by *CodeHS.com* [22]; We present the Karel variant of Fig. 4 here.

# E Symbolic Execution: Additional Details

In this section, we provide an example demonstrating how MCTS could guide the symbolic execution in generating more suitable tasks, see Fig. 25.

| (a) Initialization | (b) Search tree | (c) (1, 0, 1) | (d) (1, 1, 0) | (e) (1, 1, 1, 0, 0) |

Figure 25: Search strategy for guiding the symbolic execution; see the text below for details.

In Fig. 25b, we consider the MCTS search tree for the code $C^{out}$ from Fig. 1d obtained from the solution code of task H5 by mutation. At the top of the tree, we have a "Root" node where the initial location/direction of the agent is picked from among the available choices. Node "RU" corresponds to the block REPEATUNTIL; the child "1" from a node "RU" corresponds to unrolling of the loop (i.e., goal is false). Node "If" corresponds to the block IF; the child "1" from a node "If" corresponds to executing the code inside (i.e., pathR is true). For the purpose of this demonstration, we limited the unrolling of the loops to a maximum of 3; for our experiments, this depth is $2n(=20)$ as discussed in Appendix F. Furthermore, in this demonstration, the initial location/direction for the agent are fixed as shown in Fig. 25a; for our experiments, we consider $5 \times 4(=20)$ initial configurations which are picked by MCTS resulting in a large branching factor at the root of the tree.

Figs. 25c, 25d, and 25e illustrate the symbolic execution output for three different paths in the search tree as discussed below:

- Fig. 25c corresponds to the path ("RU":1, "If":0, "RU": 1). This path results in agent crashing the wall. As discussed in Section 4, $\mathcal{F}_{qual}(\cdot)$ score is set to 0 when $\mathcal{F}_{nocrash}(\cdot) = 0$ and hence this tree path evaluates to low $\mathcal{F}_{score}(\cdot)$ score.
- Fig. 25d corresponds to the path ("RU":1, "If":1, "RU": 0). This path results in two moves and two turns.
- Fig. 25e corresponds to the path ("RU":1, "If":1, "RU": 1, "If":0, "RU": 0). This path results in three moves and two turns, and is the higher scoring path compared to other two paths.

MCTS is extremely effective as a search strategy and quickly learns to pick paths with high scores. For the same code $C^{out}$ used in Fig. 25 above, Fig. 8a shows the temporal trends of different feature values in $\mathcal{F}_{score}$ averaged over a time window of 100 steps in our experiments. For further details, we refer the reader to [14] for an overview of the MCTS procedure, and to [15] where MCTS is used with symbolic execution to direct the exploration towards costly paths.

# F Experimental Evaluation: Additional Details and Results

This section elaborates the experimental setup and results described in Section 4. We begin by providing details on the MCTS procedure in the symbolic execution stage of our task synthesis algorithm. Furthermore, we provide insights on generating multiple tasks for a single code. We also illustrate some example output tasks which violate criteria (V) and (VI), and mutated codes which got pruned in the symbolic execution stage (see Figure 7). Our implementation of the symbolic execution stage with MCTS procedure is publicly available (see Footnote 2).

## F.1 Specification of MCTS for Single Run and Additional Results

In this section, we elaborate on the specification of MCTS, briefly discussed in Section 4. We begin by discussing choices that effect the scalability and run-time of MCTS procedure.

**Choice of initial location and direction.** When doing symbolic execution using MCTS, the procedure begins by picking an initial location and direction for the agent (see Fig. 5(b)). Given a grid-size $n$, and four initial directions (*north*, *east*, *south*, *west*) of the agent to choose from, we get a total of $4n^2$ choices for our initial configuration of the grid-puzzle. In the implementation used for generating the results, we restricted the set of initial choices to 20 (by choosing only 5 initial locations of the grid including four corners and centre, and 4 directions)—this aids in exploration by limiting the branching factor at the root of the MCTS's tree.

**Tree depth for symbolic execution.** The depth of the symbolic tree depends on the nature of the corresponding solution code. For codes without the REPEATUNTIL or WHILE constructs (H1, H2, H3, K7, K8 and K9), the tree-depth is bounded. But, for more complex codes (H4, H5, H6, and K10), the tree depth is unbounded. In our implementation, we limited the unrolling of the loops to a maximum of $2n(= 20)$. To get an insight into the complexity of the problem, we note that with 20 initial choices and a depth of $2n$, there are over 40 million leaves in the symbolic tree for codes H5 and H6 which contain conditionals nested inside REPEATUNTIL or WHILE constructs.[4] Next, we describe the details of our evaluation function for MCTS.

**Details on the evaluation function $\mathcal{F}_{\text{score}}$.** Our evaluation function $\mathcal{F}_{\text{score}}(\texttt{T}^{\text{out}}, \texttt{C}^{\text{out}}, \texttt{T}^{\text{in}}, \texttt{C}^{\text{in}}) \in [0, 1]$ measures the suitability of a generated task. A higher $\mathcal{F}_{\text{score}}$ indicates a more suitable task. We describe the elements of our evaluation function in greater detail here. We defined it in Section 4 and present it here again for completeness:

$$\mathcal{F}_{\text{score}}(\texttt{T}^{\text{out}}, \texttt{C}^{\text{out}}, \texttt{T}^{\text{in}}, \texttt{C}^{\text{in}}) = \underbrace{\mathbb{1}\Big(\mathcal{F}_{\text{qual}}(\texttt{T}_{\text{vis}}^{\text{out}}, \texttt{C}^{\text{out}}) \geq \delta_{\text{qual}}, \mathcal{F}_{\text{nocrash}}(\texttt{T}_{\text{vis}}^{\text{out}}, \texttt{C}^{\text{out}}) = 1, \mathcal{F}_{\text{nocut}}(\texttt{T}_{\text{vis}}^{\text{out}}, \texttt{C}^{\text{out}}) = 1\Big)}_{\text{2a}} \cdot$$

$$\underbrace{\Big[\alpha_1 \mathcal{F}_{\text{cov}}(\texttt{T}_{\text{vis}}^{\text{out}}, \texttt{C}^{\text{out}}) + \alpha_2 \mathcal{F}_{\text{qual}}(\texttt{T}_{\text{vis}}^{\text{out}}, \texttt{C}^{\text{out}}) + \alpha_3 \mathcal{F}_{\text{diss}}(\texttt{T}_{\text{vis}}^{\text{out}}, \texttt{T}_{\text{vis}}^{\text{in}})\Big]}_{\text{2b}} \quad (2)$$

where $\mathbb{1}$ is an indicator function and each constant $\alpha = 1/3$. It is to be noted that component 2b in Eq.2 supplies the gradients for guiding the search in MCTS. At the end of the MCTS run (containing 2 million iterations), the *best* task (i.e, the one with the highest $\mathcal{F}_{\text{score}}$ value) is picked only from the pool of generated tasks which satisfy $\mathcal{F}_{\text{cov}}(\cdot) = 1, \mathcal{F}_{\text{score}}(\cdot) > 0$. We discuss each constituent function of $\mathcal{F}_{\text{score}}$ next.

**Task quality component of evaluation function.** $\mathcal{F}_{\text{qual}}(\texttt{T}_{\text{vis}}^{\text{out}}, \texttt{C}^{\text{out}}) \in [0, 1]$ evaluates the quality and validity of $\texttt{T}^{\text{out}}$. Its is defined as a linear combination of the normalized counts of certain features of $\texttt{T}_{\text{vis}}^{\text{out}}$ when $\texttt{C}^{\text{out}}$ is executed. As certain elements differ in the two task types, HOC and Karel, we define the features differently for each. More precisely, for HOC tasks, we have:

$$\mathcal{F}_{\text{qual}}^{\text{HOC}}(\texttt{T}_{\text{vis}}^{\text{out}}, \texttt{C}^{\text{out}}) = \frac{1}{4}\Big(\frac{\#\text{moves}}{2n} + \frac{\#\text{turns}}{n} + \frac{\#\text{segments}}{n/2} + \frac{\#\text{long-segments}}{n/3}\Big)$$

where the individual features are defined as

- #moves: This refers to the count of 'moves'.
- #turns: This refers to the count of 'turns'.
- #segments: This refers to the number of consecutive sequence ($\geq 3$) of 'moves'.
- #long-segments: This refers to the number of longer consecutive sequence ($\geq 5$) of 'moves'.

For Karel tasks, we additionally have two marker based features to define the quality of the task i.e,

$$\mathcal{F}_{\text{qual}}^{\text{Karel}}(\text{T}_{\text{vis}}^{\text{out}}, \text{C}^{\text{out}}) = \frac{3}{4} \cdot \frac{1}{4}\Big(\frac{\#\text{moves}}{2n} + \frac{\#\text{turns}}{n} + \frac{\#\text{segments}}{n/2} + \frac{\#\text{long-segments}}{n/3}\Big)$$

$$+ \frac{1}{4} \cdot \frac{1}{2}\Big(\frac{\#\text{pick-markers}}{n} + \frac{\#\text{put-markers}}{n}\Big)$$

where the additional features are defined as

- #pick-markers: This refers to the count of 'pick-marker' activity.
- #put-markers: This refers to the count of 'put-marker' activity.

While we have used high-level features of $\text{T}_{\text{vis}}^{\text{out}}$ to define the quality of a task, one could also embed more specific domain knowledge in defining these features to obtain more interesting/complex tasks.

**Task dissimilarity component of the evaluation function.** $\mathcal{F}_{\text{diss}}(\text{T}_{\text{vis}}^{\text{out}}, \text{T}_{\text{vis}}^{\text{in}}) \in [0, 1]$ evaluates the visual dissimilarity of $\text{T}_{\text{vis}}^{\text{out}}$ w.r.t. $\text{T}_{\text{vis}}^{\text{in}}$. We define it as a linear combination of the dissimilarity features as follows:

$$\mathcal{F}_{\text{diss}}(\text{T}_{\text{vis}}^{\text{out}}, \text{T}_{\text{vis}}^{\text{in}}) = \frac{1}{3}\Big(\text{diss}(\text{loc} \mid \text{T}_{\text{vis}}^{\text{out}}, \text{T}_{\text{vis}}^{\text{in}}) + \text{diss}(\text{dir} \mid \text{T}_{\text{vis}}^{\text{out}}, \text{T}_{\text{vis}}^{\text{in}}) + \text{diss}(\text{grid-cells} \mid \text{T}_{\text{vis}}^{\text{out}}, \text{T}_{\text{vis}}^{\text{in}})\Big)$$

where the individual features are defined as

- $\text{diss}(\text{loc} \mid \text{T}_{\text{vis}}^{\text{out}}, \text{T}_{\text{vis}}^{\text{in}}) \in \{0, 1\}$ measures the dissimilarity in the agent's initial location in the task-puzzles $\text{T}_{\text{vis}}^{\text{out}}$ and $\text{T}_{\text{vis}}^{\text{in}}$.
- $\text{diss}(\text{dir} \mid \text{T}_{\text{vis}}^{\text{out}}, \text{T}_{\text{vis}}^{\text{in}}) \in \{0, 1\}$ measures the dissimilarity in the agent's initial direction in the task-puzzles $\text{T}_{\text{vis}}^{\text{out}}$ and $\text{T}_{\text{vis}}^{\text{in}}$.
- $\text{diss}(\text{grid-cells} \mid \text{T}_{\text{vis}}^{\text{out}}, \text{T}_{\text{vis}}^{\text{in}}) \in [0, 1]$ measures the grid-cell level dissimilarity in the task-puzzles $\text{T}_{\text{vis}}^{\text{out}}$ and $\text{T}_{\text{vis}}^{\text{in}}$. This is computed as the normalized Hamming distance w.r.t. the two grid-worlds (i.e., number of cells which are different, multiplied with a normalization factor of $\frac{2}{n^2}$).

**Deep dive into an MCTS run for Karel.** Analogous to the example provided in Section 4, we take a closer look at an MCTS run for the Karel task K10, shown in Fig. 26. Fig. 26a and Fig. 26b illustrate the improvement in various components of $\mathcal{F}_{\text{score}}$ as the number of MCTS iterations increases. Best tasks at different iterations are shown in Fig. 26c and Fig. 26d. As expected, the more the iterations, the better the tasks which are generated.

## F.2   Specification of MCTS for Multiple Runs with Diversity and Additional Results

Our task synthesis algorithm can also generate multiple tasks for a *single* code, with sufficient diversity. To achieve this, we modify the evaluation function $\mathcal{F}_{\text{score}}$ guiding the MCTS search. We introduce a diversity measure $\mathcal{F}_{\text{diversity}}$ which measures the diversity between the generated tasks. More concretely, when generating a new $(k + 1)^{\text{th}}$ task, we capture the diversity score w.r.t the tasks generated in the previous k-runs of MCTS $\{\text{T}_{\text{vis},1}^{\text{out}}, \ldots, \text{T}_{\text{vis},k}^{\text{out}}\}$.

**Modified evaluation function $\mathcal{F}_{\text{score}}$.** Our evaluation function with the new diversity component is given below with each $\alpha = 1/4$. We have $\mathcal{F}_{\text{score}}(\text{T}^{\text{out}}, \text{C}^{\text{out}}, \text{T}^{\text{in}}, \text{C}^{\text{in}}, \{\text{T}_{\text{vis},1}^{\text{out}}, \ldots, \text{T}_{\text{vis},k}^{\text{out}}\}) :=$

$$\underbrace{\mathbb{1}\Big(\mathcal{F}_{\text{qual}}(\text{T}_{\text{vis}}^{\text{out}}, \text{C}^{\text{out}}) \geq \delta_{\text{qual}}, \mathcal{F}_{\text{nocrash}}(\text{T}_{\text{vis}}^{\text{out}}, \text{C}^{\text{out}}) = 1, \mathcal{F}_{\text{nocut}}(\text{T}_{\text{vis}}^{\text{out}}, \text{C}^{\text{out}}) = 1\Big)}_{\text{3a}} \cdot$$

$$\underbrace{\Big[\alpha_1 \mathcal{F}_{\text{cov}}(\text{T}_{\text{vis}}^{\text{out}}, \text{C}^{\text{out}}) + \alpha_2 \mathcal{F}_{\text{qual}}(\text{T}_{\text{vis}}^{\text{out}}, \text{C}^{\text{out}}) + \alpha_3 \mathcal{F}_{\text{diss}}(\text{T}_{\text{vis}}^{\text{out}}, \text{T}_{\text{vis}}^{\text{in}}) + \alpha_4 \mathcal{F}_{\text{diversity}}(\text{T}_{\text{vis}}^{\text{out}}, \text{C}^{\text{out}} \mid \{\text{T}_{\text{vis},1}^{\text{out}} \ldots \text{T}_{\text{vis},k}^{\text{out}}\})\Big]}_{\text{3b}}$$

$$(3)$$

(a) Trends in $\mathcal{F}_{\text{score}}$ features

(b) Trends in $\mathcal{F}_{\text{score}}$ features capturing dissimilarity

(c) Best at 20

(d) Best at 2M

Figure 26: Illustration of a single MCTS run on $\mathtt{C}^{\text{out}}$ from Fig. 2d obtained from solution code of task K10 by mutation. (a, b) show the temporal trends of different feature values in $\mathcal{F}_{\text{score}}$ averaged over a time window of 100 steps. (c, d) show the best, i.e., highest scoring, tasks generated up to times $2 \times 10^1$ and $2 \times 10^6$ respectively.

**Diversity score of tasks** $\mathcal{F}_{\text{diversity}}$. Here, we describe the diversity component of the evaluation function. $\mathcal{F}_{\text{diversity}}(\mathtt{T}^{\text{out}}_{\text{vis}}, \mathtt{C}^{\text{out}} \mid \{\mathtt{T}^{\text{out}}_{\text{vis},1}, \ldots, \mathtt{T}^{\text{out}}_{\text{vis},k}\}) \in [0,1]$ operates on a pool of generated tasks, and computes a diversity score for the new task w.r.t the tasks in the pool. Initially, the pool of tasks generated is empty. In this case, we have $\mathcal{F}_{\text{diversity}}(\mathtt{T}^{\text{out}}_{\text{vis}}, \mathtt{C}^{\text{out}} \mid \{\}) = 1$.

After one run of MCTS, we have one task in the task pool $\{\mathtt{T}^{\text{out}}_{\text{vis},1}\}$. We define the diversity score for a subsequent task $\mathtt{T}^{\text{out}}_{\text{vis}}$ as follows. First, if $\mathtt{T}^{\text{out}}_{\text{vis}} = \mathtt{T}^{\text{out}}_{\text{vis},1}$ then we set $\mathcal{F}_{\text{diversity}}(\mathtt{T}^{\text{out}}_{\text{vis}}, \mathtt{C}^{\text{out}} \mid \{\mathtt{T}^{\text{out}}_{\text{vis},1}\}) = 0$. Otherwise, the diversity score is given by

$$\mathcal{F}_{\text{diversity}}(\mathtt{T}^{\text{out}}_{\text{vis}}, \mathtt{C}^{\text{out}} \mid \{\mathtt{T}^{\text{out}}_{\text{vis},1}\}) = \frac{1}{4}\Big(\text{diss}(\text{loc} \mid \mathtt{T}^{\text{out}}_{\text{vis}}, \mathtt{T}^{\text{out}}_{\text{vis},1}) + \text{diss}(\text{dir} \mid \mathtt{T}^{\text{out}}_{\text{vis}}, \mathtt{T}^{\text{out}}_{\text{vis},1})$$
$$+ \text{diss}(\text{grid-cells} \mid \mathtt{T}^{\text{out}}_{\text{vis}}, \mathtt{T}^{\text{out}}_{\text{vis},1}) + \text{diss}(\text{symbolic-paths} \mid \mathtt{T}^{\text{out}}_{\text{vis}}, \mathtt{T}^{\text{out}}_{\text{vis},1})\Big)$$

where the individual features are defined as

- $\text{diss}(\text{loc} \mid \mathtt{T}^{\text{out}}_{\text{vis}}, \mathtt{T}^{\text{out}}_{\text{vis},1}) \in \{0,1\}$ measures the dissimilarity in the agent's initial location in the task-puzzles $\mathtt{T}^{\text{out}}_{\text{vis}}$ and $\mathtt{T}^{\text{out}}_{\text{vis},1}$.

- $\text{diss}(\text{dir} \mid \mathtt{T}^{\text{out}}_{\text{vis}}, \mathtt{T}^{\text{out}}_{\text{vis},1}) \in \{0,1\}$ measures the dissimilarity in the agent's initial direction in the task-puzzles $\mathtt{T}^{\text{out}}_{\text{vis}}$ and $\mathtt{T}^{\text{out}}_{\text{vis},1}$.

- $\text{diss}(\text{grid-cells} \mid \mathtt{T}^{\text{out}}_{\text{vis}}, \mathtt{T}^{\text{out}}_{\text{vis},1}) \in [0,1]$ measures the grid-cell level dissimilarity in the task-puzzles $\mathtt{T}^{\text{out}}_{\text{vis}}$ and $\mathtt{T}^{\text{out}}_{\text{vis},1}$. This is computed as the normalized Hamming distance w.r.t. the two grid-worlds (i.e., number of cells which are different, multiplied with a normalization factor of $\frac{2}{n^2}$).

- $\text{diss}(\text{symbolic-paths} \mid \mathtt{T}^{\text{out}}_{\text{vis}}, \mathtt{T}^{\text{out}}_{\text{vis},1}) \in [0,1]$ measures the dissimilarity in the symbolic-paths used in the generation of $\mathtt{T}^{\text{out}}_{\text{vis}}$ and $\mathtt{T}^{\text{out}}_{\text{vis},1}$. This is computed as the normalized "edit distance" between these paths.

After we have run $k$ MCTS runs, the pool of tasks generated is populated ($\{\mathtt{T}^{\text{out}}_{\text{vis},1} \ldots \mathtt{T}^{\text{out}}_{\text{vis},k}\}$) and the diversity score of the subsequent task is computed as follows:

$$\mathcal{F}_{\text{diversity}}(\mathtt{T}^{\text{out}}_{\text{vis}}, \mathtt{C}^{\text{out}} \mid \{\mathtt{T}^{\text{out}}_{\text{vis},1}, \ldots, \mathtt{T}^{\text{out}}_{\text{vis},k}\}) = \min_{i \in [k]} \mathcal{F}_{\text{diversity}}(\mathtt{T}^{\text{out}}_{\text{vis}}, \mathtt{C}^{\text{out}} \mid \{\mathtt{T}^{\text{out}}_{\text{vis},i}\})$$

As the pool of tasks grow, it becomes more constrained to generate a task that is diverse from all the tasks in the pool. In general, there is a limit to the number of tasks per code that can be generated—eventually, the new task will have $\mathcal{F}_{\text{diversity}}$ to be 0 or $\mathcal{F}_{\text{score}}$ to be 0. As stated in Section 4, for each code $\mathtt{C}^{\text{out}}$, we generated up to 10 different visual tasks using this process.

**Illustration of output tasks using diversity.** We illustrate our diverse task generation process on both HOC and Karel tasks. Fig. 27 shows the 10 diverse tasks generated for code shown in Fig. 1d, while Fig. 28 shows the 6 diverse tasks generated for the code shown in Fig. 2d.

(a) $T_{vis,1}^{out}$  (b) $T_{vis,2}^{out}$  (c) $T_{vis,3}^{out}$  (d) $T_{vis,4}^{out}$  (e) $T_{vis,5}^{out}$

(f) $T_{vis,6}^{out}$  (g) $T_{vis,7}^{out}$  (h) $T_{vis,8}^{out}$  (i) $T_{vis,9}^{out}$  (j) $T_{vis,10}^{out}$

Figure 27: Illustration of task diversity on $C^{out}$ from Fig. 1d which was obtained from solution code of task H5 by mutation. All the 10 diverse tasks generated are shown here.

(a) $T_{vis,1}^{out}$  (b) $T_{vis,2}^{out}$  (c) $T_{vis,3}^{out}$

(d) $T_{vis,4}^{out}$  (e) $T_{vis,5}^{out}$  (f) $T_{vis,6}^{out}$

Figure 28: Illustration of task diversity on $C^{out}$ from Fig. 2d which was obtained from solution code of task K10 by mutation. First 6 diverse tasks are shown here.

## F.3 Insights into Results of Fig. 7

In this section, we provide further insights into the final results presented Fig. 7. In particular, we illustrate few limitations of the codes and tasks we generate (w.r.t task-synthesis objectives defined in Section 2):

- **Limitations of mutated codes generated**: Column 4 of Fig. 7 lists the number of codes generated by our mutation stage. However this set ($\#C_{\Delta=all}^{out}$) continues to have few semantic irregularities (as discussed in Section 3.1). Some of these irregularities are illustrated in Fig. 29. These codes are pruned by the symbolic execution stage of our algorithm, and the revised set of codes($\#C^{out}$) is listed in Column 6 of Fig. 7.

- **Tasks which violate task-synthesis objective** (V): Out of the final set of tasks generated (shown in Column 7 of Fig. 7), some of them violate objective (V). The fraction of the tasks which satisfy this particular objective are listed in Column 9 of Fig. 7. In Fig. 30 we illustrate two examples which violate the objective, in output tasks for H5 and H6.

- **Tasks which violate task-synthesis objective** $(VI)_{\delta_{mini}=1}$: Column 10 of Fig. 7 lists the fraction of the tasks generated that satisfy this particular objective, on task-minimality. In Fig. 31, we illustrate two examples of tasks which violate this minimality constraint with $\delta_{mini} = 1$.

```
def Run(){
  move
  turnRight
  move
  turnRight
  move
}
```
(a) H1: $C^{out}$

```
def Run(){
  turnLeft
  Repeat(5){
    move
    turnRight
  }
}
```
(b) H2: $C^{out}$

```
def Run(){
  Repeat(4){
    move
  }
  turnLeft
  move
  turnLeft
  Repeat(6){
    move
  }
}
```
(c) H3: $C^{out}$

```
def Run(){
  RepeatUntil(goal){
    move
    turnLeft
    move
    turnLeft
  }
}
```
(d) H4: $C^{out}$

```
def Run(){
  RepeatUntil(goal){
    turnRight
    turnRight
    move
    If(pathRight){
      turnRight
    }
  }
}
```
(e) H5: $C^{out}$

```
def Run(){
  RepeatUntil(goal){
    If(pathAhead){
      move
      turnLeft
    }
    Else{
      turnLeft
    }
  }
}
```
(f) H6: $C^{out}$

```
def Run(){
  Repeat(5){
    pickMarker
    move
    turnRight
    putMarker
  }
}
```
(g) K8: $C^{out}$

```
def Run(){
  putMarker
  While(pathAhead){
    move
    turnLeft
    move
    turnLeft
    putMarker
  }
}
```
(h) K10: $C^{out}$

Figure 29: Illustration of codes with semantic irregularities in Column $C^{out}_{\Delta=all}$ of Fig. 7. (a)–(f) show the semantic irregularities in mutated codes generated for HOC tasks. All of these codes lead to circular paths in output task, and are pruned out in the symbolic execution stage, by the $\mathcal{F}_{nocut}$ component of the $\mathcal{F}_{score}$ measure. In particular, consider the semantic irregularity presented in (g). This code corresponding to the reference task K8, has redundant marker activity. With each iteration of Repeat, the pickMarker and putMarker actions occur consecutively, leading to no change in marker activity in the output. (h) illustrates a mutated code for reference task K10 where if the While is executed more than once, putMarker activity occurs in the same location consecutively leading to a crash (as only one marker is allowed per grid-cell); if While is executed only once, the corresponding task generated has a short-cut to the goal.

(a) H5: $T^{out}_{vis}$

```
def Run(){
  RepeatUntil(goal){
    move
    If(pathRight){
      turnRight
      turnRight
    }
  }
}
```
(b) H5: $C^{out}$

(c) H6: $T^{out}_{vis}$

```
def Run(){
  RepeatUntil(goal){
    If(pathAhead){
      move
      move
      turnRight
    }
    Else{
      turnRight
    }
  }
}
```
(d) H6: $C^{out}$

Figure 30: Illustration of violation of task-synthesis objective (V) by generated output tasks for reference tasks H5 and H6. (a, b) show the irregularity in generated task ($T^{out}_{vis}$, $C^{out}$) for H5 which leads to only a straight path in the visual-puzzle. The corresponding code renders the If construct redundant. (c, d) illustrate a similar irregularity in the generated task ($T^{out}_{vis}$, $C^{out}$) for H6. Here, IfElse construct is not needed for the optimal solution.

(a) H5: $T^{out}_{vis}$

```
def Run(){
  move
  turnRight
  RepeatUntil(goal){
    move
    If(pathRight){
      turnRight
    }
  }
}
```
(b) H5: $C^{out}$

(c) H6: $T^{out}_{vis}$

```
def Run(){
  turnRight
  move
  RepeatUntil(goal){
    If(pathAhead){
      move
    }
    Else{
      turnRight
    }
  }
}
```
(d) H6: $C^{out}$

Figure 31: Illustration of violation of task-synthesis objective (V1), with $\delta_{mini} = 1$ by generated output tasks for reference tasks H5 and H6. (a, b) show the violation of task minimality, for the task ($T^{out}_{vis}$, $C^{out}$) generated for H5. In particular, the first two actions in the code shown in (b) are unnecessary. Similarly, (c, d) show the violation of task minimality, for the task ($T^{out}_{vis}$, $C^{out}$) generated for H6.

## F.4 Adding More Variability to Output Tasks

Here, we propose a few simple extensions to our task generation algorithm allowing us to add more variability in the visual puzzles. We show that high variability can be achieved through suitable post-processing and pre-processing of the generated tasks (happening after and before the symbolic execution process, respectively). We describe each of these strategies next.

**Different grid-sizes.** A simple strategy to enhance task variability is by altering the grid-size parameter $n$. Fig. 32 illustrates the tasks generated with different values of the grid-size parameter $n$. For instance, this strategy is employed in generating multiple input-output pairs for Karel tasks in *Intro to Programming with Karel* course by *CodeHS.com* [22].

(a) $\text{T}_{\text{vis}}^{\text{out}}$; $n = 10$       (b) $\text{T}_{\text{vis}}^{\text{out}}$; $n = 8$       (c) $\text{T}_{\text{vis}}^{\text{out}}$; $n = 6$

Figure 32: Tasks generated for $\text{C}^{\text{out}}$ from Fig. 1d when varying grid-size ($n$).

**Post-processing: Using distractor paths.** One of the ways to add variability to the tasks is by adding *distractor-paths*, to the generated tasks. These paths are added after the symbolic execution stage is complete, and a basic task has been generated. Fig. 33a shows the task generated without any post-processing of the output of symbolic execution, carried out on $\text{C}^{\text{out}}$ from Fig. 1d. Fig. 33b, and Fig. 33c illustrate two different post-processing patterns that yield tasks with greater variability, for the same code.

**Pre-processing: Using different initializations of grid cells.** We could also add task variability by initializing the grid-cells differently, before they are subjected to symbolic execution for task synthesis. We have a set of fixed grid-patterns, which when chosen as initializations of the grid-world yield very different looking output tasks. Fig. 34a shows the task generated for $\text{C}^{\text{out}}$ from Fig. 1d without any pre-processing on the grid-cells. Fig. 34b and Fig. 34c show two different tasks obtained using different grid-initializations, for the same code.

(symbolic input)     (symbolic output)     (visual output)

(a) No post processing

(symbolic input)     (symbolic output)     (visual output)        (symbolic input)     (symbolic output)     (visual output)

(b) Post-processing 1                 (c) Post-processing 2

Figure 33: Illustration of the (post-processing) distractor path strategy to increase task variability on tasks generated from $C^{out}$ in Fig. 1d which was obtained from solution code of task H5 by mutation. (a) illustrates the basic task obtained after symbolic execution. (b, c) show two different distractor paths added after the symbolic execution stage, yielding visually very different tasks.

(symbolic input)     (symbolic output)     (visual output)

(a) No grid-cell initialization

(symbolic input)     (symbolic output)     (visual output)        (symbolic input)     (symbolic output)     (visual output)

(b) Grid-cell initialization 1             (c) Grid-cell initialization 2

Figure 34: Illustration of the (pre-processing) grid-cell initialization strategy to increase task variability on tasks generated from $C^{out}$ in Fig. 1d which was obtained from solution code of task H5 by mutation. (a) illustrates the basic task obtained after symbolic execution. (b, c) show two different grid-cell initializations, yielding visually very different tasks.