[Reviews · NeurIPS 2020]

Review 1

Summary and Contributions: The work presents a technique for automatic generation of puzzles for block-based introductory programming courses. The goal is to generate puzzles that exercise the same concepts as a given reference (problem,solution) but on a different input. The method is based on (a) constrained mutation of a reference solution using an SMT-solver, (b) generation of a corresponding input for a new solution using MCTS over symbolic paths in the program. It was evaluated automatically on a sample of real HOS/Karel introductory tasks and also with user studies regarding the solvability and distinctness of the generated puzzles.

Strengths: + An end-to-end pipeline that generates realistic, solvable, and distinct puzzles. + Exciting combination of constraint solving, symbolic execution, and MCTS to ensure a complex set of objectives. + MCTS as a method to constrain symbolic execution could be used for many input generation tasks in the ML4Code community. + Comprehensive and systematic evaluation both on automatically verifiable metrics and on the effects of generated problems on user engagement.

Weaknesses: - Apart from the fundamental parts of MCTS, nothing is really learned. While the application is clearly relevant to the NeurIPS community, much of the solution is not. - Due to the complexity of the overall pipeline that has to be presented in 8 pages, most of the technical meat of the paper is left to the Appendix. - Some important parts are not elaborated even in the Appendix. - The broader impact of this technique on the students' concept learning and progression is unclear.

Correctness: The claims and methodology are sound, but some parts of the methodology are not explained. The details of evaluating the constraints (V)-(VI) in L240-242 are unclear. What is "a random sample of up to 100 tasks?" Is it 100/task or overall? Are the denominators of the fractions in Fig.7 cols 9-11 comparable? L243: How exactly do you apply delta debugging?

Clarity: The paper is mostly well-written, but many technical details in the main body of the paper are deferred to the Appendix (understandably, the page limit is tight for such an expansive work). Some are not expanded sufficiently even there, namely: a) The actual combination of MCTS and symbolic execution, the most interesting technical part of the work. I would appreciate: - an example of unrolling a specific program and collecting input constraints during SE, and - a formal definition of MCTS applied to such an unrolling. b) Delta debugging to verify minimality of two puzzles. The notation is a bit inconsistent. It mostly follows a "Hungarian-like" format of V_name where V is the same for variables of the same type (e.g. δ for distance thresholds). I was confused with it initially, but the pattern grew on me as I kept reading. However, this pattern is broken for e.g. T_in, T_store, and T_size.

Relation to Prior Work: Related: Code editing has been used for *feedback generation* in introductory programming [1], but not for task generation. The SOTA for program synthesis in Karel is probably [2] rather than the works cited in L54. It approaches human performance although does not meet the experts' level of brevity. The Ω/ω mapping between the program space and sketch space resembles abstract interpretation. The connection could be emphasized explicitly. [1] Rolim, R., Soares, G., D'Antoni, L., Polozov, O., Gulwani, S., Gheyi, R., Suzuki, R. and Hartmann, B., 2017, May. Learning syntactic program transformations from examples. In 2017 IEEE/ACM 39th International Conference on Software Engineering (ICSE) (pp. 404-415). IEEE. [2] Chen, X., Liu, C. and Song, D., 2018, September. Execution-guided neural program synthesis. In International Conference on Learning Representations.

Reproducibility: No

Additional Feedback: The main concern with this work is its relevance to NeurIPS. The application is clearly relevant, as it combines program synthesis, constraint satisfaction, and intelligent tutoring, all well-established in the AI literature. However, the solution is almost entirely symbolic – it combines SMT solving, symbolic execution, and delta debugging over programs. The only probabilistic component is MCTS, and even there (as far as I understand) it is vanilla MCTS without learned policies. The paper has many interesting contributions, but none are related to machine learning. I think the work is valuable and many of its parts are of interest to the intelligent tutoring community. However, I wonder if it's better suited for PLDI, OOPSLA, or AAAI/IJCAI. This prevents me from giving it a slightly higher score. The user studies are well executed and very much appreciated for the paper. For an important educational application, relying on automatic measurements would not genuinely evaluate how appropriate the generated problems actually are for the target audience. That said, the evaluation doesn't exactly reach the motivational goal set in the introduction. While they measure *appropriateness* of the generated problems and demonstrate that they don't hinder solving ability *too much*, they don't evaluate their *usefulness*. In other words, the evaluation relies on the core assumption that *exercising a programmatic concept is equivalent to solving problems with similar enough programmatic solutions*, and this assumption needs to be tested in practice. Prior works on different educational domains (e.g. Ahmed et al., Polozov et al.) relied on the same assumption, so I don't blame the authors for following it. However, investigating how well this syntactic restriction relates to actual concept learning would be an interesting area of future work. In general, design of curricula in education or levels in gaming is nonlinear and does not necessarily repeat the same procedural concept on different inputs [3]. [3] Butler, E., Smith, A.M., Liu, Y.E. and Popovic, Z., 2013, October. A mixed-initiative tool for designing level progressions in games. In Proceedings of the 26th annual ACM symposium on User interface software and technology (pp. 377-386). Unclear question: How much knowledge (constraints, sketches, objective functions, etc.) is pre-specified universally, and what needs to be specified per domain/per puzzle?


Review 2

Summary and Contributions: This paper presents a new technique to automatically synthesize practice tasks for block-based programming problems given a reference task. The key idea of the technique is to first perform mutations in the program solution that satisfy certain set of constraints. The second step of the solution is to use MCTS for generating task descriptions (visual inputs) given the mutated code in the first step. The technique is evaluated on Hour of Code and Karel problems, and the synthesized problems are evaluated using a user study that shows that the approach performs in a comparable fashion to the expert designed problems.

Strengths: + Interesting problem domain of automatically generating practice programming problems + Nice two level decomposition of the solution strategy to first generate mutated code and then generate corresponding visual inputs + Detailed user study to evaluate the generated problems

Weaknesses: - The sketch constraints for performing code mutations seem a bit domain-specific - Manual effort to specify domain constraints for Sketch generation and MCTS - No baseline comparisons for mutation and symbolic execution

Correctness: Yes, the claims and methods seem correct. The empirical methodology is also sound, but it is missing some baseline comparisons.

Clarity: Yes, the paper is well written. Some details about the F_qual and F_diss could be moved from supplementary material to the main text.

Relation to Prior Work: Yes, the previous works are described in details and compared with.

Reproducibility: Yes

Additional Feedback: Overall I quite liked the problem domain of automatically generating programming problems for the block-based environments. The strategy for decomposing the problem into two steps -- first generating code performing sketch-based mutations and the second step of using MCTS to generate visual inputs, is quite elegant. The user study demonstrates usefulness of the proposed technique. Some of the sketch constraints and MCTS objective function seemed a bit too domain-specific for Karel style problems. One question I had was about the generality of the constraints. If one were to use the same technique for other programming environments (say introductory Python problems), how much additional manual work will be needed to get the technique to work? From Figure 7, the number of programs obtained after applying the first constraint do not seem very large in number (max 132M for K10). I was wondering whether one can simply enumerate all of these programs and evaluate the programs to check for the constraints rather than trying to solve them using z3. Would such an approach not scale? Similarly for the second part of visual input generation, I was wondering why z3 can't be used here to also generate inputs? One approach could be to use a symbolic encoding to generate inputs for different paths in the code and then filter out the ones that violate quality, dissimilarity, crash, or no change constraints. Also, how well would a random search for input generation compare? Many programming tasks require multiple input-output examples to specify different corner cases. Even for Karel, many programs might require worlds of different grid shapes to handle different branch conditions. Would it be straightforward to extend the MCTS algorithm to generate multiple inputs for a given program? Currently, the approach assumes that the control flow of the program is a good measure for defining problem concept and only performs mutations at the block level. I was curious if there was also some evidence that it actually corresponds to the perceived difficulty by students as well? I was also wondering whether some of the manual constraints that are currently explicitly encoded could also be learnt in a more automated fashion from possibly a corpus of problems of similar difficulty level. For the user study results for the SAME task, weren't the users provided with the code solutions for T_in? I was curious why the fraction of solved tasks in this case was 1.00. Update post author response: Thanks for the response. It would be good to add details about F_score and F_qual from appendix to main text, and also the ablation experiments with simpler enumerative approaches. Also some discussions about how the approach can be extended to handle generating multiple I/O examples and why control flow could be a good measure of problem complexity would be helpful.


Review 3

Summary and Contributions: This paper attempts to create/synthesise visual programming tasks by methods that automatically generate candidate referents based on similarity. Their algorithm is composed of first mutated existing sets, then performing symbolic execution over them, using MCTS to guide search in the execution tree. Effectiveness of the algorithm is measure with empirical experiments on several block-based games as well as user studies with humans. Update: I think this is a good-enough contribution to warrant an accept and I hope the authors take into account the notes on more detailed evaluation, because I think that would be helpful for anyone who wants to use this/have work that spins off of this. Updating my score slightly..

Strengths: 1. This is an interesting problem that ties to many other relevant problems in machine learning. e.g., in general, learning to generate candidate sets of referents based on characteristics of the objects, as well as user/pragmatic outputs, will be largely helpful in the multi-agent emergent communication community as well. 2. The paper is well written and all the components of the algorithm and its working are clearly explained. 3. It's nice that there is a user study component in the evaluation that assess how this works.

Weaknesses: 1. The visual dissimilarity measures (Likert scale) could possible be improved/made better (since it seems like this affect performance as well)

Correctness: The experimental setup is sound and explained in detail.

Clarity: The paper is written well and all experimental details and tables and figures are adequately explained.

Relation to Prior Work: This is well-positioned in the literature.

Reproducibility: Yes

Additional Feedback:


Review 4

Summary and Contributions: This paper is about generating new programming tasks structurally similar to existing ones for pedagogical purposes. The domain is "block-based programming", where students write programs that control an agent in a grid-world, based on a visual specification for what the program should do; for example, the student is provided an initial state and a final state for the world, and tasked with writing the program that would transform the initial state into the final state. When provided with an existing specification, the algorithm in the paper produces new specifications which can be solved with programs that are similar to the solution for the original specification. By doing this, we can automatically generate many new interesting tasks that exercise similar concepts as an existing one, enabling students to practice their understanding of the concepts. The paper presents a formalization of the task of generating new specifications, an algorithm based on symbolic execution and an SMT solver, and an empirical evaluation comparing against baseline methods in a user study.

Strengths: The paper defines and motivates an interesting problem, presents a compelling algorithm for the problem, and shows that it works well with a user study. The methods used in the paper (applying a SMT solver, using MCTS, etc) are not novel on their own, but they make intuitive sense and appear to be a good fit for the problem in the empirical evaluation. I liked that the paper has a formal presentation of the problem and then uses it meaningfully as part of the solution (with the SMT solver). In other papers, it is not uncommon that the formalisms are mostly for expository purposes, and not directly relevant in the proposed method. The paper only evaluated the method on one domain, but most of the approach appears generic enough that it would be easily adapted to other programming languages (for example, teaching how to use regular expressions).

Weaknesses: In Section 3.2, I felt that Monte Carlo Tree Search was an unnecessarily opaque method for the guidance of the symbolic execution. The advantage is that it can work with many kinds of constraints and cost functions, but it is not very computationally efficient as many evaluations are needed. For the evaluation functions used in this paper, would it be possible to use a more efficient method tailored for them? Also, how would MCTS compare with other methods for searching the space of paths, like evolutionary algorithms? I was glad to see the empirical user study for the method, but I felt that the baselines used could have been more varied. In particular, with an ablation study where important components of the overall system are removed or modified, we can see more clearly their importance to achieving the end result. If the study budget allows, I would encourage the authors to consider more user studies that show the importance of e.g. different evaluation functions used in the MCTS. I was also a bit surprised to see that the study participants were recruited using Mechanical Turk, as the programming classes are targeted at K-12 students but Mechanical Turk requires workers to be at least 18 years old. The users participating in the study would have different demographics than the intended users for the system, so the latter may also behave differently in ways that invalidate conclusions drawn from the user study. Finally, there is not much learning involved in the paper, compared to most other NeurIPS submissions.

Correctness: I didn't notice any issues with the correctness.

Clarity: The paper was clear and easy to understand.

Relation to Prior Work: The paper claims to define a new task, and there is no related work section. However, the paper briefly discuses other prior work for problem generation [3, 10, 24, 30] in the introduction.

Reproducibility: Yes

Additional Feedback: In Definition 4 (line 89), what if the task can be solved with distinct programs that have different nesting structures? I didn't understand the utility of this definition.

[Author Response · NeurIPS 2020]

We thank the reviewers for their valuable suggestions. Please find our answers for each reviewer (**R**) below.

[**R1, R2: Additional technical details in the paper and appendices**] As suggested by Reviewer 1, we will provide
a detailed example of the combination of MCTS and symbolic execution, and add details on Delta debugging. As
per Reviewer 2, we will move the details of $\mathcal{F}_{\text{score}}$ and $\mathcal{F}_{\text{qual}}$ to the main paper. Especially, if the paper is accepted,
the extra page in the final submission will allow us to incorporate these additional details in the main paper.

[**R1, R2: Pre-specification of domain knowledge**] To answer Reviewer 1's question, the domain knowledge is mostly
universal to both HOC and Karel tasks—a few differences arise primarily because Karel's DSL is slightly different (also,
see $\mathcal{F}_{\text{qual}}^{\text{HOC}}$ and $\mathcal{F}_{\text{qual}}^{\text{Karel}}$ in L570 and L576 respectively). To answer Reviewer 2, the manual effort in designing the Sketch
constraints and cost function was actually minimal, compared to the effort required to manually generate a large pool
of new tasks. As suggested, it is an interesting direction to automatically learn the constraints and cost function from a
corpus. To apply our methodology to other programming environments (e.g., Python problems), one should first establish
the specification of tasks/problems: it would be easier to extend our methodology with tasks specified as I/O pairs.

[**R1, R4: Lack of learning aspects in the solution**] We thank the reviewers for appreciating the importance of the
problem and its relevance to the NeurIPS community. Regarding concerns on the lack of learning aspects in the solution,
we believe that the richness of the problem setting and obtained results will spur interesting follow-up works requiring
more complex learning-based solutions. One exciting direction, as hinted by Reviewer 1, is to learn a policy to guide
the MCTS procedure (instead of running vanilla MCTS). Another important direction, as suggested by reviewers,
is to automatically learn the constraints and cost function from a human-generated pool of problems.

Beyond the application domain of programming education, our methodology can be used for generating large-scale
training datasets consisting of tasks and solution codes with desirable characteristics—this can be potentially useful for
researchers in the NeurIPS community, for example, to improve the state of the art on neural program synthesis.

[**R1, R4: Students' concept learning and additional user studies**] We generally agree that more elaborate studies will
be needed to fully reach the motivational goal of teaching K-12 students, and evaluate the long term impact on students'
concept learning. After the publication of this work, it is quite conceivable to perform larger-scale studies through colla-
borations with existing educational platforms towards this goal. As suggested by Reviewer 4, we will also consider doing
additional user studies varying different components. Nevertheless, the results demonstrate the benefits of our approach.

[**R2, R4: Baseline comparisons for mutation and symbolic execution**] We will include additional ablation studies
in the updated paper. As suggested by Reviewer 2, we ran the mutation stage by enumerating the programs within size
constraints and then post-checking other constraints without Z3. This implementation leads to a run-time increase by a
factor of 10 to 100 for different tasks. So, Z3 seems to be very effective by jointly considering all the constraints.

As a search method, although MCTS seems computationally expensive, the actual run-time and memory footprint of
an MCTS run depends on the unique traces explored (i.e., unique symbolic executions done)— this number is typically
much lower than the number of iterations, also see Footnote 1 below L571. Furthermore, MCTS is extremely effective
as a search strategy, see Fig. 8a. Reviewer 2 asked for a comparison of MCTS with a random search: Considering
the MCTS output in Fig. 8c, 8d, to obtain a comparable evaluation score through random search, the corresponding
number of unique symbolic executions required is at least 10 times more than executed by MCTS.

[**R1: Related work and clarifications**] We thank the reviewer for pointing out the missing references. Regarding
L240–242, we analyzed a random sample of 100 outputs per reference task and we will clarify this in the updated paper.
We note that it is possible to increase this number but manually checking properties (V), (VI) is time-consuming for
tasks with nested structures. Regarding Delta debugging in L243, we note that the task minimality statistics reported
in columns 10, 11 of Fig. 7 are based on human evaluation; Delta debugging was applied primarily to aid the human
evaluation. We will add full details in the paper.

[**R2: Multiple I/O pairs and user study results**] Our methodology can be easily extended to multiple I/O pairs
for Karel by adapting techniques from Appendix E.2 designed for generating diverse tasks. The reviewer raises an
interesting question on the correlation between program control flow and problem concept: This seems to be the case
for similar expert-generated problems in platforms like *Code.org* and *CodeHS.com*; also see our response above on
additional user studies. SAME's performance is below 1.00 possibly because participants overlooked the solution of
Step 1 unaware they will be receiving the same task in Step 2, and the app did not allow them to go back to Step 1.

[**R3: Likert scale for visual dissimilarity**] Indeed, it would be interesting to understand the sensitivity of user study
results w.r.t. the scale definition. Another possibility is to use pairwise comparisons in eliciting user evaluations.

[**R4: Clarification on Definition 4**] If a task $T^{\text{out}}$ can be solved by multiple distinct programs, then all those programs
should have the same nesting structure as $C_{\text{struct}}^{\text{in}}$. This strict definition of conceptual similarity is important as students
do not have the solution code $C^{\text{out}}$. Also, we refer the reviewer to property (V) in L106 and Figure 29 in Appendix E.3.

[Meta-Review · NeurIPS 2020]

Overall, reviewers liked how the paper combined constraint solving, symbolic execution and MCTS to solve the problem. The main reservation was some concern about relevance to the venue given the lack of any learning in the approach. However, reviewers believed that both the problem and the techniques are of interest to people in the community, for example working on ML-guided program synthesis.